# EqGINO: Equivariant Geometry-Informed Fourier Neural Operators for 3D PDEs

**Sungwon Kim** [1] **Juho Song** [2] **Seungmin Shin** [3] **Guimok Cho** [3] **Sangkook Kim** [3] **Chanyoung Park** [1 4]

## Abstract

Deep learning surrogates for 3D Partial Differential Equations (PDEs) often fail to generalize across geometric transformations because they depend heavily on specific coordinate systems. While equivariant networks offer a solution, they typically rely on local operations in the spatial domain, making the global receptive field—essential for PDE dynamics—computationally expensive. Conversely, Fourier Neural Operators (FNOs) efficiently capture global interactions, yet establishing 3D equivariance within them remains impractical due to the prohibitive cost of spectral group convolutions. To bridge this gap, we introduce EqGINO, a geometrically robust framework that enforces isotropy in the spectral domain. By design, EqGINO guarantees exact equivariance to the discrete symmetries inherent to the discretized computational domain. Beyond this discrete guarantee, our structural prior enables effective generalization to arbitrary continuous orientations even with a limited number of SE(3)-transformed training samples. Consequently, our method robustly models coordinate-invariant physical laws on complex irregular 3D geometries. Our code is available at this URL.

## 1. Introduction

Solving Partial Differential Equations (PDEs) provides a powerful framework for simulating the physical laws governing the real world. These simulations play a critical role in various fields, including structural engineering (Whalen et al., 2021), material science (Zhang et al., 2018), industrial

manufacturing (Pfaff et al., 2020), and robotics (Makoviychuk et al., 2021). However, solving PDEs is inherently challenging. While numerical methods (Klocke et al., 2002; Felippa, 2004) based on discretization have traditionally been the standard, they suffer from high computational costs and slow execution times, severely limiting their applicability in real-time and large-scale scenarios. Consequently, deep learning-based surrogate models (Li et al., 2020a; 2023; Wu et al., 2024; Kim et al., 2025) have emerged as a promising alternative, capable of rapidly approximating PDE solutions and overcoming the bottlenecks of traditional solvers.

Despite this potential, a fundamental disparity persists between the intrinsic coordinate-invariance of physical laws and the coordinate-dependent nature of current neural models (Park & Kang, 2024; Li et al., 2023; 2020a;b; Wu et al., 2024). Governing equations, such as the Navier-Stokes, remain valid regardless of the observer's reference frame. In contrast, many state-of-the-art models heavily rely on Cartesian coordinate inputs to facilitate training. While providing such positional information stabilizes optimization for fixed tasks, it inevitably biases the model toward canonical orientations, causing it to overfit to the training coordinate system rather than learning the underlying physical dynamics. Consequently, these models struggle to generalize when initial conditions or geometries undergo rigid transformations—such as rotation or translation—severely compromising their reliability in dynamic, real-world environments.

While equivariant architectures offer a promising solution to mitigate coordinate dependency, particularly those grounded in geometric deep learning, they were not originally designed to address the unique challenges of PDE solving (Satorras et al., 2021; Trang et al., 2024). That is, their reliance on local message passing inherently restricts their receptive fields, rendering them less suitable for modeling the long-range, global interactions fundamental to physical dynamics (Li et al., 2018; Alon & Yahav, 2020). On the other hand, Fourier Neural Operators (FNOs) (Li et al., 2020a) operating in the spectral domain offer distinct advantages, such as the inherent ability to capture global correlations. Furthermore, the spectral domain provides a natural representation for differential operators, transforming complex convolutions and derivatives in physical space into simple pointwise multiplications (Li et al., 2020a; Trefethen, 2000).

[1]Graduate School of Data Science, KAIST, Daejeon, Republic of Korea [2]Mathematical Sciences, KAIST, Daejeon, Republic of Korea [3]LG Electronics, Pyeong-taek, Republic of Korea [4]Industrial & Systems Engineering, KAIST, Daejeon, Republic of Korea. Correspondence to: Chanyoung Park <cy.park@kaist.ac.kr>.

*Proceedings of the 43rd International Conference on Machine Learning*, Seoul, South Korea. PMLR 306, 2026. Copyright 2026 by the author(s).

This property facilitates precise modeling of PDE dynamics with reduced discretization errors compared to conventional local difference schemes (Trefethen, 2000; Boyd, 2001). However, establishing 3D equivariance within this domain remains underexplored. While a recent study has addressed this in 2D (Helwig et al., 2023), extending such methods to 3D entails prohibitive computational costs due to the complexity of 3D spectral group convolutions. Given the practical demand for 3D simulations, there is a pressing need to investigate efficient, equivariant modeling in the 3D spectral domain for large-scale applications.

To address these challenges, we propose the **Eq**uivariant **G**eometry-**I**nformed Fourier **N**eural **O**perator (**EqGINO**), a framework designed to bridge the gap between 3D equivariance and computational efficiency in 3D spectral modeling. While leveraging the structural backbone of GINO (Li et al., 2023)—which enables global spectral analysis on arbitrary geometries via the Fast Fourier Transform (FFT)—we fundamentally redesign the core processing units, originally based on the GNO (Li et al., 2020b) and FNO (Li et al., 2020a), into two standalone equivariant modules: the *EqGNO* and *EqFNO*. Specifically, the EqGNO serves as a geometric operator for processing unstructured geometric inputs under global rigid transformations, whereas the EqFNO functions as a general-purpose equivariant backbone for 3D grid-based learning tasks requiring symmetry preservation. By integrating these modules, EqGINO guarantees exact equivariance to the discrete SE(3) subgroups inherent to the discretized computational domain, ensuring physical consistency without sacrificing efficiency.

At the core of EqGINO lies a novel 3D spectral layer that achieves equivariance through an *Orbit-based Weight Sharing* strategy. By sharing weights among frequency modes belonging to the same radial orbit, we guarantee isotropy in the spectral domain. This strategy not only reduces parameter complexity compared to the vanilla FNO–shifting from $\mathcal{O}(K^3)$ to a linear scaling $\mathcal{O}(K)$ with respect to the spectral resolution $K$ per dimension–but also serves as a strong structural prior. Beyond the discrete guarantee, this prior enables the model to effectively generalize to arbitrary continuous orientations, accelerating convergence to superior performance even with a limited number of SE(3)-transformed training samples. Consequently, EqGINO attains equivariance while retaining the expansive receptive field without the prohibitive costs of 3D group convolutions by design, thereby adapting robustly to unseen geometric transformations. Our contributions are summarized as follows:

- We propose EqGINO, a unified framework that integrates 3D equivariance into the global receptive field induced by the spectral convolution, enabling robust generalization across irregular 3D geometries.

- We introduce *Orbit-based Weight Sharing* to enforce spec-

tral isotropy, reducing the parameter complexity of spectral convolution from volumetric to linear scaling while guaranteeing exact discrete equivariance.

- We demonstrate that EqGINO guarantees exact consistency on discrete orientations by design and seamlessly extends to arbitrary continuous orientations even with limited SE(3)-transformed data, outperforming state-of-the-art baselines.

## 2. Related Work

### 2.1. Neural Operators for PDEs

Effective PDE surrogate models must capture global interactions, particularly in physics systems such as incompressible fluid dynamics, where local changes necessitate instantaneous responses in distant regions. Consequently, the ability to resolve such long-range dependencies is a fundamental prerequisite for the physical validity of any solver.

To address this, the Fourier Neural Operator (FNO) (Li et al., 2020a) leverages the Fast Fourier Transform (FFT) (Cooley & Tukey, 1965) to capture global correlations efficiently, though it is restricted to regular grids. To handle irregular geometries, the Graph Neural Operator (GNO) (Li et al., 2020b) operates on arbitrary meshes but incurs high costs for dense graphs. GINO (Li et al., 2023) unifies these by employing GNOs to project irregular data onto a latent grid for FNO processing, supporting scalable global modeling on complex domains.

Parallel to this, Transformer-based (Vaswani et al., 2017) models approximate global integral operators (Kovachki et al., 2023), with architectures like Galerkin Transformer (Cao, 2021) and GNOT (Hao et al., 2023) adopting linear attention to reduce computational costs. Transolver (Wu et al., 2024) further optimize this paradigm, allowing scalable learning of intrinsic physical states on massive geometries.

However, the heavy dependence on absolute coordinates causes these methods to be fragile to rigid transformations, resulting in physically inconsistent predictions.

### 2.2. Equivariance in Physical and Spectral Domains

While SE(3)-equivariant models such as EGNN (Satorras et al., 2021) and EMNN (Trang et al., 2024) successfully address coordinate dependency, their reliance on local message passing renders them inherently inefficient for PDE surrogate modeling. Because PDE dynamics often involve global correlations, capturing these effects via local propagation incurs prohibitive computational costs. Even attempts to mitigate this by incorporating long-range edges (e.g., T-EMNN (Kim et al., 2025)) offer only partial improvements, falling short of achieving a truly global receptive field.

In contrast, the spectral domain inherently offers a global

receptive field, making it ideal for capturing long-range interactions. However, 3D equivariant spectral models remain underexplored; G-FNO (Helwig et al., 2023) becomes computationally prohibitive in 3D due to group convolutions, while EGNO (Xu et al., 2024) focuses on temporal rather than spatial equivariance. Consequently, achieving a scalable, spatially equivariant spectral model for 3D PDEs remains an open challenge.

## 3. Preliminaries

### 3.1. Problem Formulation

Let $\mathcal{A}$ and $\mathcal{U}$ be separable Banach spaces of functions defined on a bounded domain $D \subset \mathbb{R}^3$. We consider a system of partial differential equations (PDEs) governing a physical system, formally posed as a boundary value problem:

$$\begin{aligned} \mathcal{L}(a, u) = 0 & \quad \text{in } D, \\ \mathcal{B}(u) = 0 & \quad \text{on } \partial D, \end{aligned} \tag{1}$$

where $a \in \mathcal{A}$ represents input parameters (e.g., geometry, initial conditions), and $u \in \mathcal{U}$ denotes the solution field (e.g., velocity or pressure). Here, $\mathcal{L}$ is the differential operator encoding the physical laws (e.g., Navier-Stokes equations), and $\mathcal{B}$ is the boundary operator enforcing constraints such as Dirichlet or Neumann conditions on the boundary $\partial D$. Our objective is to learn a surrogate operator $G_\theta : \mathcal{A} \to \mathcal{U}$, parameterized by $\theta$, that approximates the solution operator $G^\dagger$, such that $G_\theta(a) \approx u$ where $u$ satisfies Eq. (1).

### 3.2. SE(3)-Equivariance in Physical Systems

Physical laws are inherently independent of the observer's coordinate frame. Formally, let $\mathcal{G} = SE(3)$ be the Special Euclidean group in 3D, consisting of rotations $R \in SO(3)$ and translations $\mathbf{t} \in \mathbb{R}^3$. The group action of $g = (R, \mathbf{t}) \in \mathcal{G}$ on the spatial domain is defined as the affine transformation $g \cdot x = Rx + \mathbf{t}$. In homogeneous coordinates, this corresponds to a matrix multiplication by $T_g \in \mathbb{R}^{4 \times 4}$. Correspondingly, the group action on a input point cloud $\mathcal{P} = \{y_j\}_{j=1}^N$ transforms the set into $\{Ry_j + \mathbf{t}\}_{j=1}^N$.

Consider a feature field $v$ belonging to a function space $\mathcal{H}$. We define the group action $T_g^* : \mathcal{H} \to \mathcal{H}$ via the pullback operator associated with a representation $\rho$ of $\mathcal{G}$:

$$[T_g^* v](x) := \rho(g^{-1}) v(g^{-1} \cdot x). \tag{2}$$

Here, $g^{-1} = (R^T, -R^T \mathbf{t})$. The specific form of $\rho$ depends on the geometric nature of the space. We denote the representations as $\rho_{in}$ and $\rho_{out}$ for the input space $\mathcal{A}$ and the output space $\mathcal{U}$, respectively. For a scalar field, the values remain invariant under transformation, which implies a trivial representation ($\rho(g) = I$). Throughout this paper, we focus on scalar fields to simplify the notation. Thus, the

reduced operation $[T_g^* v](x) = v(R^{-1}(x - \mathbf{t}))$ accounts for both the rotational orientation and the translational shift of the underlying domain.

Formally, a neural operator $G_\theta$ is equivariant if $G_\theta(T_g^* a) = T_g^* G_\theta(a)$ for all $g \in \mathcal{G}$ and inputs $a \in \mathcal{A}$. Satisfaction of this property is crucial for the accurate prediction of physical systems across arbitrary coordinate systems.

## 4. Methodology

### 4.1. Architecture Overview

The overall architecture integrates two proposed equivariant primitives—*EqGNO* and *EqFNO*—into a unified pipeline consisting of three stages: (i) the EqGNO encoder $\mathcal{E}$ that lifts irregular geometric information into latent representations on a regular grid; (ii) consecutive EqFNO layers $\mathcal{K}_l$ that efficiently compute global interactions retaining equivariance through isotropic spectral convolution; (iii) and the EqGNO decoder $\mathcal{D}$ that reconstructs the target physical field on the original point cloud.

$$G_\theta = \mathcal{D} \circ \mathcal{K}_L \circ \cdots \circ \mathcal{K}_1 \circ \mathcal{E}. \tag{3}$$

While our architecture adopts the structural backbone of GINO (Li et al., 2023), we fundamentally redesign each module to enforce SE(3)-equivariance. Crucially, this modularity allows each component to function as an independent, general-purpose module: EqFNO serves as an equivariant backbone for 3D grid-based learning tasks requiring symmetry preservation (e.g., volumetric physics simulation), while EqGNO acts as a geometric operator for processing unstructured geometric inputs under global rigid transformations.

We detail our SE(3)-equivariant GNO encoder and decoder (EqGNO) in Sec. 4.2, followed by the SE(3)-equivariant FNO (EqFNO) in Sec. 4.3.

### 4.2. SE(3)-Equivariant Graph Neural Operator block

The Graph Neural Operator (GNO) (Li et al., 2020b) is effective for mapping data between irregular meshes and structured latent grids. However, the original formulation parameterizes the kernel using absolute Cartesian coordinates. Since these coordinate values vary under rotation, the standard GNO fails to produce consistent features for the same geometric object in different orientations. To overcome this limitation, we adapt the GNO to be SE(3)-equivariant by employing a rotation-invariant kernel strategy.

**Encoder.** Our EqGNO encoder $\mathcal{E}$ transforms the irregular point cloud $\mathcal{P}$ into a latent representation $v_0$ at each uniform regular grid point $x^{grid}$ by approximating a local kernel integral with a Riemann sum: $v_0(x^{grid}) \approx \sum_{j=1}^M \kappa(x^{grid}, y_j) \mu_j$, where $\{y_j\}_{j=1}^M \subset B_r(x^{grid})$ de-

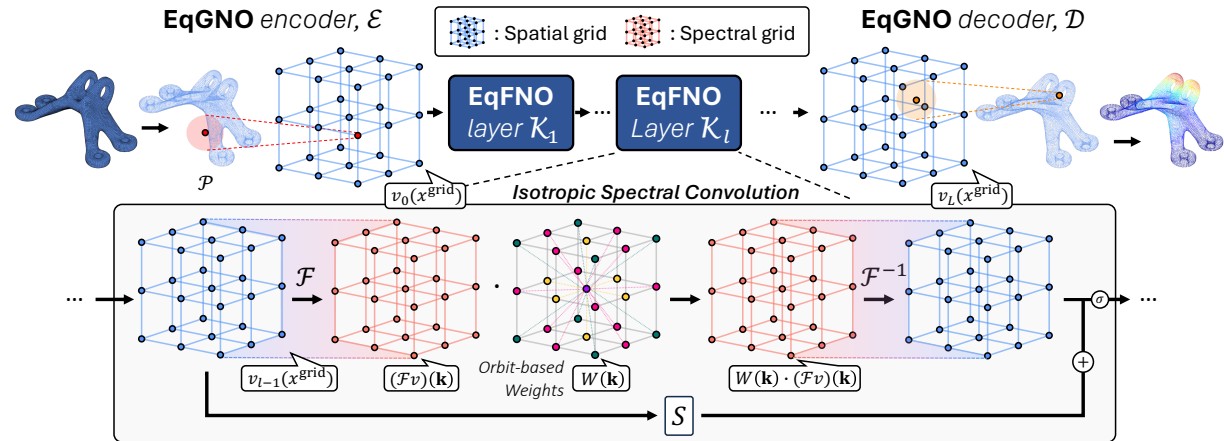

*Figure 1.* **Overview of EqGINO.** The encoder $\mathcal{E}$ aggregates local geometric information from irregular inputs $\mathcal{P}$ and maps it to regular spatial grid. Within the Fourier layers $\mathcal{K}_l$, orbit-based weights $W(\mathbf{k})$ mix the Fourier-transformed feature vectors $(\mathcal{F}v)(\mathbf{k})$ at each Fourier mode $\mathbf{k}$, ensuring equivariance. The decoder $\mathcal{D}$ projects the updated features $v_L(x^{grid})$ enriched with global context back to the point cloud to predict the physical field $u$ (e.g., deflection, pressure). All modules preserve equivariance by construction.

notes the input points sampled randomly within a local ball of radius $r$ centered at $x^{grid}$, and $\mu_j$ is the quadrature weight compensating for irregular density.

To enforce SE(3)-equivariance, we replace coordinate-dependent inputs with scalar invariants. Specifically, the kernel $\kappa$ takes the relative distance $\|x^{grid} - y_j\|$ and $\|x^{grid} - \bar{y}\|$ as inputs where $\bar{y}$ denotes the centroid of $\mathcal{P}$:

$$\kappa(x^{grid}, y_j) = \phi_\theta \left( \|x^{grid} - y_j\|, \|x^{grid} - \bar{y}\| \right). \quad (4)$$

This design ensures that the kernel captures both the local relative spatial structure and the global radial context without being sensitive to the object's orientation. Detailed proofs are provided in Appx. E.

**Decoder.** Our EqGNO decoder $\mathcal{D}$ projects the final grid representation $v_L$ back to the point cloud $Y_{out} = \{y^{out}\}$ in the physical space. Serving as the inverse of the encoding process, it reconstructs the predicted physical quantity $u$ at each target coordinate $y^{out}$ via kernel integration over neighboring grid points:

$$u(y^{out}) \approx \sum_{x_j^{grid} \in \mathcal{N}(y^{out})} \kappa_{dec}(y^{out}, x_j^{grid}) v_L(x_j^{grid}) \mu_j. \quad (5)$$

Here, $\kappa_{dec}$ is a learnable kernel parameterized by the relative distance $\|y^{out} - x_j^{grid}\|$ and $\|y^{out} - \overline{x^{grid}}\|$, analogous to the encoder. By relying solely on this rotation-invariant quantity, the decoder preserves the SE(3)-equivariance.

### 4.3. SE(3)-Equivarariant Fourier Neural Operator block

Even though EqGNO ensures SE(3)-equivariance for the geometric transformations, the intermediate processing—specifically the Fourier Neural Operator (FNO)

blocks—must also preserve this property to guarantee end-to-end equivariance. In this section, we extend the FNO (Li et al., 2020a) to be SE(3)-equivariant (EqFNO). We first formulate the Fourier layer and then theoretically derive the conditions required to embed equivariance.

**Fourier Layer.** We define the spatial convolution operator $\mathcal{L} : \mathcal{H}_{in} \rightarrow \mathcal{H}_{out}$ acting on an input function $v_{in} \in \mathcal{H}_{in} = L^2(D; \mathbb{R}^{d_{in}})$ with a kernel function $\kappa : D \times D \rightarrow \mathbb{R}^{d_{out} \times d_{in}}$ as follows:

$$(\mathcal{L}v_{in})(x) = \int_D \kappa(x, y) v_{in}(y) dy. \quad (6)$$

We impose periodic boundary conditions on the domain $D$ and that $\kappa$ is translation-invariant as in FNO. This allows the Convolution Theorem to diagonalize the convolution operator in the spectral domain. Let $\mathcal{F}$ and $\mathcal{F}^{-1}$ denote the Fourier transform and its inverse. The action of the operator $\mathcal{L}$ can be expressed as:

$$(\mathcal{L}v)(x^{grid}) = \mathcal{F}^{-1} \left[ W(\mathbf{k}) \cdot (\mathcal{F}v)(\mathbf{k}) \right](x^{grid}), \quad (7)$$

where $\mathbf{k} \in \mathbb{Z}^3$ represents the Fourier mode (spectral grid point). Here, $W(\mathbf{k})$ denotes the learnable weight, which corresponds to the Fourier coefficient of the kernel $\kappa$ for each Fourier mode, i.e., $W(\mathbf{k}) = (\mathcal{F}\kappa)(\mathbf{k}) \in \mathbb{C}^{d_{out} \times d_{in}}$.

By incorporating the channel-mixing spectral convolution, we define the complete Fourier Layer $\mathcal{K} : \mathcal{H}_{in} \rightarrow \mathcal{H}_{out}$ as:

$$\mathcal{K}_l(v_{l-1})(x^{grid}) = \sigma(S_l v_{l-1}(x^{grid}) + (\mathcal{L}v_{l-1})(x^{grid})) \quad (8)$$

For layers $l = 1, \ldots, L$, the feature updates iteratively as $v_l = \mathcal{K}_l(v_{l-1})$. The final output $v_L$ then serves as the input to the EqGNO decoder $\mathcal{D}$. Here, $S \in \mathbb{R}^{d_{out} \times d_{in}}$ denotes a learnable weight matrix. The skip connection $Sv(x)$ adjusts

for non-periodic boundary effects in the spatial domain (Li et al., 2020a). A non-linear activation $\sigma$ mixes frequency components and facilitates the learning of non-linear solution operators.

**Analysis of Fourier Neural Operator.** A fundamental motivation for applying Fourier representations is the inherent compatibility between the Fourier transform and spatial rotations. To analyze the equivariance properties, we first recall the behavior of scalar fields under rotation in the spectral domain:

**Lemma 4.1** (Spectral Rotation Property). *Let $f \in L^2(\mathbb{R}^3)$ be a scalar field and $\mathcal{T}_R$ be the rotation operator defined by $[\mathcal{T}_R f](x) = f(R^{-1}x)$. The Fourier transform satisfies:*

$$\widehat{\mathcal{T}_R f}(\mathbf{k}) = \hat{f}(R^{-1}\mathbf{k}). \tag{9}$$

This lemma formally establishes that a rotation in the spatial domain induces a coherent rotation of the Fourier modes in the spectral domain (Proof provided in Appx. F).

The Fourier transform exhibits intrinsic rotational equivariance, as shown in Lemma 4.1. However, the FNO fails to preserve this property because it comprises learnable spectral weights. Specifically, the spectral weights $W(\mathbf{k})$ in the convolution layer operate as unconstrained parameters and do not adhere to the rotational transformation law of Fourier modes. This incompatibility with rotation-induced shifts in the spectral domain undermines the equivariance inherited from the Fourier transform. Such unconstrained parameterization causes a violation of SO(3)-equivariance, as the network fails to differentiate between orientation changes and fundamentally different objects.

**EqFNO: Isotropic Spectral Convolution.** We now identify the necessary and sufficient conditions required to achieve rotation equivariance in the spectral domain. To guarantee that our EqFNO architecture respects the geometric symmetries of the physical system, the spectral convolution layer must enforce equivariance. Formally, this requirement mandates that the convolution operator $\mathcal{L}$ commute with the pullback operator $T_g^*$ in the spectral domain: $\mathcal{L} \circ T_g^* = T_g^* \circ \mathcal{L}$. That is, for any rotation $R \in SO(3)$ and any input feature field $h$, the following condition must hold:

$$\mathcal{F}\left(\mathcal{L}(T_g^* h)\right)(\mathbf{k}) = \mathcal{F}\left(T_g^*(\mathcal{L}h)\right)(\mathbf{k}). \tag{10}$$

By expanding both sides via the properties of the induced spectral action, we derive the necessary condition on the learnable weight matrix $W(\mathbf{k})$. Notably, translation equivariance is inherently guaranteed by the properties of convolution; the spatial translation $\mathbf{t}$ corresponds to a phase shift $e^{-2\pi i \langle \mathbf{k}, \mathbf{t} \rangle}$ in the spectral domain. Since we exclusively use coordinate-independent features—meaning both input

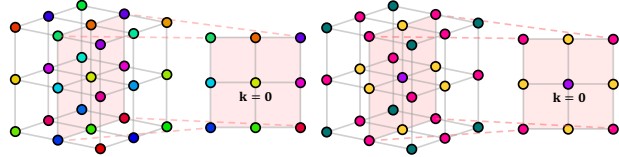

*(a)* Anisotropic weights      *(b)* Isotropic weights

*Figure 2.* Visualization of weight parameters within FNO layers. Identical colors denote shared parameters. (a) Anisotropic weights in the vanilla FNO. (b) Isotropic weights in the proposed EqFNO.

and output fields consist of rotation-invariant scalar channels—the weight constraint simplifies to:

$$W(R\mathbf{k}) = W(\mathbf{k}). \tag{11}$$

This implies that the learnable weights must remain invariant under any rotation of the Fourier mode $\mathbf{k}$. In other words, the weight $W(\mathbf{k})$ depends solely on the frequency magnitude $\|\mathbf{k}\|_2$, rather than its direction. We provide a proof of the equivariance including translation in Appx. G.

To enforce the constraint in Eq. (11) explicitly, we introduce an *Orbit-based Weight Sharing* mechanism. We define an orbit $\mathcal{O}_r$ as the set of Fourier modes with the same magnitude: $\mathcal{O}_r = \{\mathbf{k} \in \mathbb{Z}^3 \mid \|\mathbf{k}\|_2 \approx r\}$. Instead of learning independent weights for each $\mathbf{k}$, we assign a single shared weight parameter $w_r$ to all modes within the same orbit $\mathcal{O}_r$ as shown in Fig. 2b.

**Necessity of Full-FFT for Equivariance.** We explicitly employ the Full-FFT rather than the Real-FFT (RFFT). RFFT reduces computational redundancy by storing only a spectral half-space, exploiting the Hermitian symmetry of real-valued signals ($\hat{f}(-\mathbf{k}) = \overline{\hat{f}(\mathbf{k})}$). However, this compression scheme is structurally incompatible with rotation equivariance. A rotation can map a Fourier mode from the stored half-space to the omitted region, necessitating reconstruction via complex conjugation. Since complex conjugation is an anti-linear operation, it violates the complex-linear assumption of the spectral convolution layer. We avoid this conflict entirely by adopting Full-FFT. Collectively, these structural constraints ensure that the model strictly satisfies rotation equivariance. Details are given in Appx. H.

***Remark 1. How is the parameter complexity of Full-FFT handled?*** The implementation of the Full-FFT doubles the size of the learnable spectral tensor compared to the RFFT. However, the weight sharing constraint significantly mitigates this overhead by decreasing the total count of independent parameters. Quantitatively, for the spectral resolution $K^3$, our approach reduces parameter complexity from $\mathcal{O}(K^3)$ to $\mathcal{O}(K)$, as the number of orbits scales linearly.

***Remark 2. How is the computational overhead of Full-FFT addressed?*** While weight sharing reduces parameter complexity, the increased computational load (FLOPs)

of Full-FFT presents a challenge. To mitigate this, we enforce a block-diagonal structure on the spectral weight tensor (Xie et al., 2017). This constraint restricts the linear transformation to independent subspaces, decomposing the original dense weight matrix into $G$ smaller, disjoint sub-matrices. For each $c \in \{1, \ldots, G\}$ and $\mathbf{k}$, the sub-weight matrix $[W(\mathbf{k})]_c$ resides in $\mathbb{C}^{\frac{d_{out}}{G} \times \frac{d_{in}}{G}}$. Thus, the operation replaces a single large matrix multiplication with $G$ independent smaller matrix multiplications. The approximation for the channel-mixing convolution computation cost becomes $G \cdot (\frac{d_{out}}{G} \cdot \frac{d_{in}}{G} \cdot N) = \frac{d_{out}d_{in}N}{G}$, where $N$ denotes the total number of Fourier modes. While this constraint limits channel interaction, it decreases the FLOPs of the channel mixing by a factor of $G$, effectively offsetting the extra computation required for the full set of Fourier modes relative to RFFT.

With $G = 2$, the computational cost aligns with the RFFT baseline. For larger grouping factors ($G > 2$), we translate the efficiency gain into higher per-axis resolution of the spectral grid ($K$). This adjustment compensates for weight sharing constraints; a higher $K$ yields a larger number of distinct radial orbits, effectively increasing independent learnable parameters. Consequently, this approach restores the model's expressive capacity while maintaining the number of learnable parameters comparable to the RFFT baseline.

### 4.4. SE(3)-Equivariant Local Basis

In the previous section, the proposed orbit-based weight sharing mechanism guarantees equivariance for predicting scalar tasks such as von Mises stress prediction by enforcing the isotropic condition $W(R\mathbf{k}) = W(\mathbf{k})$. However, the extension of this framework to vector field prediction, such as displacement $\mathbf{u} \in \mathbb{R}^3$, presents a fundamental challenge, as described in Appx. G.

To resolve this challenge, we propose a geometric strategy that reformulates the vector prediction task as a scalar regression problem within our efficient isotropic backbone. Rather than directly predicting global vector components, the model infers the projection coefficients $\{\alpha, \beta, \gamma\}$ of the target vector onto an SE(3)-equivariant local basis $\{\mathbf{e}_1, \mathbf{e}_2, \mathbf{e}_3\}$. Consequently, the final vector field $\mathbf{v}$ is reconstructed via the linear combination: $\mathbf{v} = \alpha\mathbf{e}_1 + \beta\mathbf{e}_2 + \gamma\mathbf{e}_3$. This approach extends the model's capability to the prediction of SE(3)-equivariant vector fields. Further implementation details and the specific methodologies for constructing local bases for each dataset are provided in Appx. I.

**Discrete Equivariance and Generalization.** While our theoretical formulation guarantees SE(3)-equivariance, the reliance on regular grids for the FFT inherently restricts exact equivariance to the Octahedral rotation group ($O$). However, our architecture, grounded in SE(3)-equivariance, serves as a strong structural prior, enabling EqGINO to

effectively generalize to arbitrary SE(3) transformations and significantly outperform baselines even with limited SE(3)-transformed training data, as empirically demonstrated in the subsequent experiments.

## 5. Experiment

**Datasets.** We evaluate our method on three distinct 3D physics benchmarks covering both fluid dynamics and structural mechanics. Details are provided in Appx. B.

- **Fluid Dynamics (AhmedBody & ShapeNetCar):** The *AhmedBody* (Ahmed et al., 1984; Li et al., 2023) represents turbulent flow regimes around parameterized vehicle shapes, predicting the vector-valued *wall shear stress* and four scalar quantities: *pressure*, *turbulent kinetic energy*, *turbulent viscosity*, and *specific dissipation rate (omega)*. The *ShapeNetCar* (Umetani & Bickel, 2018; Chang et al., 2015) involves external flow simulations over diverse vehicle geometries, estimating the surface *pressure* field.

- **Structural Mechanics (DeepJEB):** The *DeepJEB* (Hong et al., 2025) dataset follows the principles of linear elasticity in solid mechanics. The prediction task aim to estimate the structural response of mechanical components under external loads and torques, specifically the vector-valued *deflection* and the scalar *von Mises stress*.

**Evaluation Protocol.** We assess SE(3)-equivariance using three protocols (Table 1 and 2). For all rotated settings, we transform both input geometry and vector-valued quantities (e.g., velocity, forces) to maintain physical consistency. **(a) In-Distribution (Canonical → Canonical):** Training and evaluation are performed on canonically aligned data to establish a performance baseline within a fixed reference frame. **(b) Zero-Shot Generalization (Canonical → Discrete Rotations):** The canonically trained model is evaluated on samples rotated by $90°$ multiples (the $O$ group). This tests the model's intrinsic geometric robustness without prior exposure to rotated data. **(c) Continuous Generalization (Rotated → Continuous Rotations):** Training and evaluation are conducted using continuous SE(3) augmentations to assess robustness against arbitrary orientations.

**Baselines.** We benchmark our method against a diverse set of baselines, including point cloud-based methods (PointNet (Qi et al., 2017a), PointNet++ (Qi et al., 2017b), PointDeepONet (Park & Kang, 2024), Transolver (Wu et al., 2024), GINO (Li et al., 2023)) and mesh-based GNNs (MeshGraphNet (Pfaff et al., 2020), EGNN (Satorras et al., 2021), EMNN (Trang et al., 2024), T-EMNN (Kim et al., 2025)).

To analyze the trade-off between generalization and expressivity, we evaluate both non-equivariant and equivariant models. Non-equivariant baselines fully leverage canonical information (e.g., coordinates) to maximize in-distribution

*Table 1.* Performance comparison for **(a)** In-Distribution and **(b)** Zero-Shot Generalization ($O$ group). We evaluate *relative $L_2$ error*. Best results are **bolded**, and second-best are underlined.

| Model | (a) Train: Canonical → Test: Canonical | | | | | | | | (b) Train: Canonical → Test: Rotated (Discrete) | | | | | | | |
|---|---|---|---|---|---|---|---|---|---|---|---|---|---|---|---|---|
| | AhmedBody | | | | | ShapeNet | DeepJEB | | AhmedBody | | | | | ShapeNet | DeepJEB | |
| | WS | Press | Kin | Visc | Omega | Press | Defl | Stress | WS | Press | Kin | Visc | Omega | Press | Defl | Stress |
| **Non-Equivariant Models** | | | | | | | | | | | | | | | | |
| PointNet | 0.4270 | 0.5119 | 0.3758 | 0.1136 | 0.0414 | 0.1433 | 0.1738 | 0.3864 | 0.9547 | 1.1866 | 0.6203 | 0.1978 | 0.0451 | 1.4418 | 4.5059 | 1.3424 |
| PointNet++ | 0.4495 | 0.5457 | 0.3699 | 0.0992 | 0.0377 | 0.1570 | 0.2028 | 0.4296 | 1.1461 | 0.9105 | 0.7071 | 0.1607 | 0.0487 | 1.7209 | 2.7154 | 1.3535 |
| MeshGraphNet | 0.3473 | 0.3842 | 0.2416 | 0.0927 | 0.0224 | 0.6291 | 0.4147 | 0.5088 | 1.0072 | 1.1495 | 0.6560 | 0.1638 | 0.0449 | 0.6291 | 8.5468 | 2.0466 |
| GINO | 0.1987 | 0.1666 | **0.1454** | 0.0584 | 0.0146 | 0.1610 | **0.1113** | 0.4026 | 0.6238 | 0.5631 | 0.4265 | 0.1296 | 0.0385 | 1.4950 | 2.3194 | 1.1266 |
| PointDeepONet | 0.7231 | 0.8906 | 0.9731 | 0.1722 | 0.0319 | 0.2015 | 0.1870 | 0.5221 | 1.6741 | 1.4503 | 1.6456 | 0.2111 | 0.0661 | 1.4337 | 5.2359 | 1.3851 |
| Transolver | **0.1286** | 0.2763 | 0.1922 | 0.0833 | 0.0159 | **0.1194** | 0.1615 | **0.3735** | 0.7946 | 0.5189 | 0.5066 | 0.1350 | 0.0469 | 1.6632 | 1.5885 | 1.0422 |
| **Equivariant Models** | | | | | | | | | | | | | | | | |
| EGNN | 0.9950 | 0.9184 | 1.0231 | 0.1797 | 0.0339 | 0.7070 | 0.8423 | 0.5636 | 0.9950 | 0.9184 | 1.0231 | 0.1797 | 0.0339 | 0.7070 | 0.8423 | 0.5636 |
| EMNN | 0.9772 | 0.9309 | 0.9256 | 0.1751 | 0.0319 | 0.2509 | 0.8028 | 0.6155 | 0.9772 | 0.9309 | 0.9256 | 0.1751 | 0.0319 | 0.2509 | 0.8028 | 0.6155 |
| Transolver* | 0.7458 | 0.6620 | 0.4690 | 0.1350 | 0.0361 | 0.9295 | 0.4192 | 0.5236 | 0.7458 | 0.6620 | 0.4690 | 0.1627 | 0.0361 | 0.9295 | 0.4192 | 0.5236 |
| T-EMNN | 0.7295 | 0.7844 | 0.8316 | 0.1674 | 0.0281 | 0.1831 | 0.3200 | 0.4204 | 0.7295 | 0.7844 | 0.8316 | 0.1674 | 0.0281 | 0.1831 | 0.3200 | 0.4204 |
| **EqGINO*** | 0.2078 | 0.1737 | 0.1513 | 0.0613 | 0.0157 | 0.1878 | 0.1644 | 0.3877 | 0.2078 | 0.1737 | 0.1513 | 0.0613 | 0.0157 | 0.1878 | **0.1644** | 0.3877 |
| **EqGINO** | 0.1958 | **0.1637** | 0.1489 | **0.0581** | **0.0137** | 0.1773 | 0.1710 | 0.3853 | **0.1958** | **0.1637** | **0.1489** | **0.0581** | **0.0137** | 0.1773 | 0.1710 | **0.3853** |

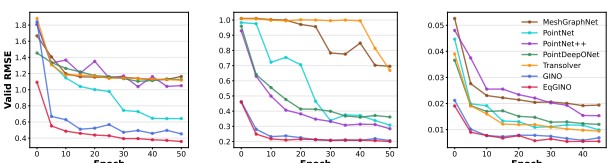

*Figure 3.* Visualization of *deflection* predictions on the *DeepJEB* dataset. Note that all models were trained exclusively on the canonical trainset. The rows display input geometries in different orientations: **(Top)** Canonical input; **(Bottom)** Input rotated by $180°$. Contours indicate the magnitude of deflection, while the error maps depict the absolute difference in magnitude (|Ground Truth − Prediction|). Qualitative analysis and extended visualizations for additional baselines are provided in Appx. K and L.1.

*Figure 4.* Validation RMSE of models trained with discrete rotation augmentation, evaluated on the test set under identical discrete rotations. (**Left**) *AhmedBody (Wall Shear Stress)*, (**Middle**) *ShapeNetCar (Press)*, (**Right**) *DeepJEB (Deflection)*.

expressivity, whereas equivariant baselines are restricted to geometric invariants (e.g., curvature, dot products) to ensure robust generalization across transformations.

For the state-of-the-art Transolver (Wu et al., 2024), we provide a comprehensive evaluation by comparing two versions: (1) the vanilla Transolver with standard coordinate features (*Transolver*), and (2) a custom SE(3)-equivariant variant (*Transolver\**). This modified version relies solely on scalar invariants—such as the distance from the point of force application and average face area—and incorporates our proposed local basis method for vector-valued predictions. By contrasting two variants, we can quantify the impact of explicit coordinate features on model performance. Further implementation details are provided in Appx. A

## 5.1. In-Distribution Performance on Canonical Poses

In Table 1a, we present a comprehensive performance comparison against all baseline models on the canonical test dataset. We report results for EqGINO under two configurations representing distinct resource constraints: *EqGINO\** ($G = 2, K = 32$), which aligns with the computational cost of the baseline *GINO*, and *EqGINO* ($G = 4, K = 40$), which maintains an equivalent parameter budget to *EqGINO\** while prioritizing higher spectral resolution. Here, $G$ corresponds to the number of disjoint channel groups within the block-diagonal structure of the weight tensor, as detailed in Sec. 4.3.

The results demonstrate that EqGINO achieves performance competitive with non-equivariant state-of-the-art models (*Transolver*, and *GINO*), despite the inherent restrictions on input node features required to enforce equivariance. Furthermore, EqGINO significantly outperforms mesh-based equivariant baselines (*EGNN*, *EMNN*, and *T-EMNN*), a distinct advantage attributed to the global receptive field provided by spectral analysis. In contrast, the SE(3)-equivariant variant of Transolver (i.e., *Transolver\**), which relies solely on scalar features without explicit coordinate information, exhibits a marked degradation in performance. This substantial drop exposes the difficulty of achieving high performance without coordinates, a challenge our spectral ap-

*Table 2.* Performance comparison for Continuous Generalization ($SE(3)$ group). We evaluate *relative $L_2$ error*. Best results are **bolded**, and second-best are underlined.

| | Train: Rotated (Continuous) → Test: Rotated (Continuous) | | | | | | | |
|---|---|---|---|---|---|---|---|---|
| | AhmedBody | | | | | ShapeNet | DeepJEB | |
| Model | WS | Press | Kin | Visc | Omega | Press | Defl | Stress |
| **Non-Equivariant Models** | | | | | | | | |
| PointNet | 0.3561 | 0.3405 | 0.2351 | 0.1018 | 0.0385 | 0.2366 | 0.2144 | 0.3895 |
| PointNet++ | 0.3675 | 0.3884 | 0.2937 | 0.0923 | 0.0312 | 0.2417 | 0.2706 | 0.4397 |
| MeshGraphNet | 0.6028 | 0.4127 | 0.2754 | 0.1265 | 0.0217 | 0.5842 | 0.5097 | 0.5143 |
| GINO | 0.2779 | 0.2110 | 0.1747 | 0.0660 | 0.0178 | 0.1813 | 0.1584 | 0.4197 |
| PointDeepONet | 0.6336 | 0.6888 | 0.6720 | 0.2124 | 0.0409 | 0.3375 | 0.3744 | 0.7615 |
| Transolver | 0.3349 | 0.4222 | 0.3219 | 0.1239 | 0.0359 | 0.3347 | 0.2169 | **0.3658** |
| **Equivariant Models** | | | | | | | | |
| EGNN | 0.8619 | 0.8181 | 0.6989 | 0.2025 | 0.0297 | 0.6543 | 0.8375 | 0.5923 |
| EMNN | 0.8844 | 0.7669 | 0.6865 | 0.2116 | 0.0302 | 0.2522 | 0.8182 | 0.6553 |
| Transolver* | 0.9430 | 0.6424 | 0.4549 | 0.1610 | 0.0347 | 0.9273 | 0.4014 | 0.5123 |
| T-EMNN | 0.6994 | 0.6203 | 0.5967 | 0.1581 | 0.0266 | 0.1795 | 0.3045 | 0.4244 |
| **EqGINO*** | 0.2247 | **0.1775** | **0.1491** | 0.0651 | **0.0160** | 0.1658 | **0.1553** | 0.3693 |
| **EqGINO** | **0.2087** | 0.1848 | 0.1494 | **0.0612** | 0.0163 | **0.1560** | 0.1621 | 0.3668 |

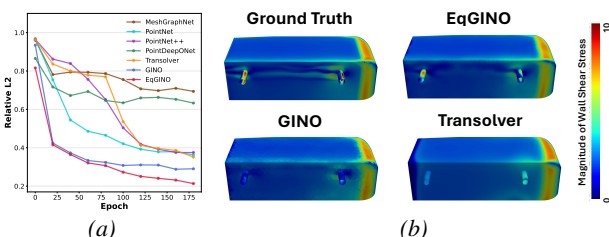

*(a)*                   *(b)*

*Figure 5.* Evaluation of rotation robustness on the *AhmedBody* dataset. (**a**) Relative $L_2$ error curves for models trained and validated under random rotation augmentation with continuous angles $\theta \in [0, 360°]$. (**b**) Magnitude of the predicted *Wall Shear Stress* vectors for a test sample rotated by $58.7°$ about the $x$-axis. Extended visualizations are provided in Appx. L.3

proach effectively overcomes.

## 5.2. Zero-Shot Generalization to Discrete Rotations

After training on data in canonical positions, we evaluate the models using randomly rotated test data. Specifically, we apply rotations with angles selected from $0°, 90°, 180°, 270°$ around a randomly chosen axis ($x$, $y$, or $z$). As shown in Table 1b, our method, EqGINO, outperforms all baselines, including other equivariant models. Fig. 3 visualizes the predicted 3D deflection on the DeepJEB dataset under varying input geometric orientations. When the input shape is aligned with the canonical position (top row), errors are generally small across models. Fourier-based models (EqGINO, GINO) particularly demonstrate superior capability in modeling torsional loads, which requires capturing global behaviors. Conversely, when the input is rotated (e.g., upsidedown), non-equivariant baselines fail significantly as they are overfitted to the canonical orientation (bottom row). These results highlight EqGINO's unique ability to maintain precise global modeling while adhering to equivariance.

**Can Data Augmentation Replace Equivariance?** To investigate whether data augmentation can substitute for architectural equivariance, we trained non-equivariant base-

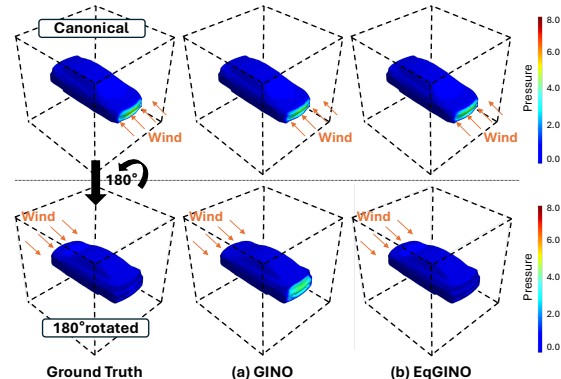

*Figure 6.* Visualization of pressure predictions on the *ShapeNetCar* dataset. All models were trained on the canonical trainset. The rows display input geometries in different orientations: (**Top**) Canonical input; (**Bottom**) Input rotated by $180°$. Orange arrows indicate the wind direction. Colors represent the pressure magnitude. Extended visualizations for additional baselines are provided in Appx. L.2.

lines using data augmented with rotations identical to the test distribution. As illustrated in Fig. 4, while augmentation improves general robustness, it fails to achieve uniform generalization across all orientations. In contrast, EqGINO benefits from intrinsic structural equivariance, consistently outperforming augmented baselines while requiring significantly fewer training samples. This confirms that learned equivariance falls short of exact architectural guarantees.

## 5.3. Sample-Efficient Generalization to Continuous Rotations

While EqGINO guarantees exact equivariance within the discrete subgroup ($O$), real-world scenarios often involve arbitrary rotations in the continuous domain ($SO(3)$). To extend our model's robustness to these continuous variations, we incorporate rotational augmentation sampled from the full $SO(3)$ group during training. In Table 2, EqGINO consistently outperforms all baselines on the SE(3)-transformed test dataset. Interestingly, as shown in Fig. 5a, we observe that EqGINO generalizes to these unseen continuous orientations significantly faster and achieves superior performance compared to the baselines. Specifically, Fig. 5b demonstrates that EqGINO accurately captures detailed physical behaviors on complex geometries—such as the turbulent regions around pillar structures—even when the input is arbitrarily rotated (e.g., $58.7°$). This suggests that the inherent discrete equivariance provides a strong structural foundation, which enables the model to learn the remaining continuous symmetries more efficiently than models lacking this geometric prior.

## 5.4. Qualitative Analysis of Coordinate Dependency

We conduct a qualitative evaluation to verify the presence of coordinate dependency in the baseline models. Fig. 6 visu-

alizes the pressure field predictions of GINO and EqGINO, both trained exclusively on canonical poses. As observed in the Ground Truth, aerodynamic pressure is concentrated on the front bumper due to the incoming wind. However, GINO fails to adapt to the rotated scenario (Fig. 6a); it erroneously predicts high pressure on the rear bumper simply because rotation places it in the spatial region previously occupied by the front bumper. This suggests that coordinate-dependent models overfit to the absolute coordinate system of the training data rather than learning the physical relation between geometry and fluid dynamics. In contrast, EqGINO (Fig. 6b) consistently identifies the high-pressure region on the actual front bumper regardless of its spatial location. This demonstrates that our model relies on intrinsic geometric features rather than extrinsic coordinates, avoiding the overfitting issues inherent in coordinate-dependent approaches.

Additional results on discretization-convergence and ablation studies are provided in the Appx. J.

## 6. Conclusion

In this paper, we presented EqGINO, a geometrically robust framework for 3D PDE surrogate modeling. By fundamentally redesigning the core processing units into standalone equivariant modules, our approach achieves spectral isotropy via an *Orbit-based Weight Sharing* strategy, significantly reducing parameter complexity while ensuring physical consistency. Empirically, EqGINO demonstrates superior robustness across diverse 3D physics benchmarks. While state-of-the-art solvers struggle to generalize across geometric transformations due to their reliance on absolute coordinates, our architecture effectively mitigates this dependency without compromising performance in canonical settings. Future work will focus on strictly enforcing continuous equivariance beyond grid limitations and reducing the overall computational cost to accommodate million-scale industrial geometries.

## Acknowledgements

This work was supported by LG Electronics and the National Research Foundation of Korea (NRF) grants funded by the Korea government (MSIT) (RS-2024-00335098, RS-2022-NR068758).

## Impact Statement

This paper presents work whose goal is to advance the field of Machine Learning. There are many potential societal consequences of our work, none which we feel must be specifically highlighted here.

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

# A. Implementation Details and Experimental Settings

In this section, we provide a comprehensive description of the hyperparameters and training configurations used in our experiments to ensure reproducibility.

## A.1. General Training Configurations

To ensure a fair comparison, we maintained consistent optimization settings across all baseline models and our proposed method. We utilized the Adam optimizer for all experiments. The initial learning rate was tuned via grid search within the candidate set $\{1 \times 10^{-4}, 5 \times 10^{-4}, 1 \times 10^{-3}\}$. Due to the substantial memory requirements associated with processing high-density 3D meshes and point clouds, a batch size of 1 was employed. Consequently, Batch Normalization (Ioffe & Szegedy, 2015) was deemed unsuitable due to unstable statistics. We therefore substituted it with Group Normalization (Wu & He, 2018) specifically for the point-based models: PointNet, PointNet++, and PointDeepONet. Regarding model capacity, all baseline architectures, with the exception of the GINO baseline, were configured with a hidden dimension of 128. For our proposed EqGINO and the GINO baseline, the hidden dimension was set to 64.

## A.2. Model-Specific Architectures

For all baseline models, we utilized the official source codes whenever available to ensure reproduction of the original methods. Specifically, we implemented PointDeepONet and TEMNN from scratch, as their official implementations were not publicly accessible.

For point-based baselines (PointNet, PointNet++, and PointDeepONet), we adapted the architectures for regression tasks on 3D fields. Notably, for the torsional loading case in PointDeepONet, the model cannot directly process global direction vectors. To address this, we explicitly calculated the force magnitude and direction corresponding to the torque at each node and incorporated these as input node features.

Among Graph Neural Networks, MeshGraphNet explicitly utilized absolute coordinate inputs. In contrast, EGNN and EMNN operated without absolute coordinate inputs. TEMNN employed a specialized invariant coordinate system constructed intrinsically from the mesh structure. For the Transformer-based baseline, we customized the vanilla Transolver architecture to enforce SE(3)-equivariance (denoted as *Transolver*** in Table 1). The input node features for this model were specifically designed to represent the local geometry of the nodes. Specifically, for the **AhmedBody** and **ShapeNetCar** datasets, we computed invariant geometric descriptors using the `trimesh` library: *vertex degrees*, representing local connectivity, and *vertex defects* (angle defects), representing discrete Gaussian curvature calculated as the deviation of the sum of incident face angles from $2\pi$. For the **DeepJEB** dataset, we incorporated physics-informed geometric features, including the distance from fixed points, the distance from load application points, the dot product between the force vector and the surface normal, and the force magnitude.

GINO and our proposed EqGINO were constructed with 4 Fourier Neural Operator (FNO) layers. The integration radius for the Graph Neural Operator (GNO) was tuned specifically for each dataset. For AhmedBody ($r = 0.1$) and ShapeNetCar ($r = 0.15$), the geometries were scaled to the range $[-1, 1]$. For DeepJEB, the radius was set to $r = 10.0$, and the data was processed at its original scale. A critical distinction of our EqGINO architecture is its independence from explicit coordinate inputs, relying solely on intrinsic geometric information, whereas the GINO typically requires coordinate features. Regarding resolution configurations, the GINO baseline utilized $K = 32$. For EqGINO, we evaluated two configurations: $K = 32$ with a grouping factor of $G = 2$, and $K = 40$ with a grouping factor of $G = 4$.

*Table 3.* Summary of node features. Coordinate-dependent features serve non-equivariant baselines, while invariant features ensure SE(3)-invariance for our model.

| Dataset | Coord-Dependent *(Non-Equi Only)* | Invariant Geometric & Physical Features *(Equivariant & All Models)* |
|---|---|---|
| **AhmedBody** | $\mathbf{x}$ (Abs. Coord), $\mathbf{u}_{in}$ (Inlet Velocity) | **Geom:** Vertex degrees, Vertex defects (discrete Gaussian curvature, angle deviation from $2\pi$). **Phys:** Projected Inlet Velocity ($\mathbf{u}_{in} \cdot \mathbf{n}$). |
| **ShapeNetCar** | $\mathbf{x}$ (Abs. Coord), $\mathbf{u}_{in}$ (Inlet Velocity) | **Geom:** Vertex degrees, Vertex defects (same as above). **Phys:** Projected Inlet Velocity ($\mathbf{u}_{in} \cdot \mathbf{n}$). |
| **DeepJEB** | $\mathbf{x}$ (Abs. Coord) | **Geom:** Geodesic dist. from fixed boundary points, Geodesic dist. from force application points. **Phys:** Projected Force ($\mathbf{F} \cdot \mathbf{n}$), Force Magnitude ($|\mathbf{F}|$). |

## B. Dataset Specifications

The fluid dynamics simulations for both the AhmedBody and ShapeNetCar datasets are based on solving the Reynolds-Averaged Navier-Stokes (RANS) equations, which represent the time-averaged form of the incompressible Navier-Stokes equations. These equations take the form of the momentum balance and continuity equations:

$$\rho(\mathbf{u} \cdot \nabla)\mathbf{u} = -\nabla p + \mu \nabla^2 \mathbf{u} + \mathbf{f},$$
$$\nabla \cdot \mathbf{u} = 0 \tag{12}$$

where $\mathbf{u}$ is the velocity field, $p$ is the pressure, $\rho$ is the fluid density, $\mu$ is the dynamic viscosity, and $\mathbf{f}$ represents body forces (Ferziger et al., 2019). Both datasets consists of fluid dynamics simulations computed under the same RANS framework and are selected to assess the ability of EqGINO to efficiently handle large-scale simulations (*AhmedBody*) while maintaining robust generalization across complex and diverse geometric configurations (*ShapeNetCar*).

**AhmedBody.** The *AhmedBody* dataset (Ahmed et al., 1984) represents a canonical problem in vehicle aerodynamics. The dataset comprises simulations of parametrized car-like shapes subject to turbulent airflow. The model predicts the vector-valued *wall shear stress* alongside four scalar physical quantities: *pressure, turbulent kinetic energy, turbulent viscosity,* and *specific dissipation rate (omega)*. All target physical fields are evaluated on the surface mesh of the geometry. We split the dataset into 413 simulations for training, 45 simulations for validation, and 50 simulations for testing.

**ShapeNetCars.** To complement the AhmedBody benchmark and assess the robustness of EqGINO to geometric diversity, we additionally adopt the *ShapeNetCar* dataset. This benchmark derives from the car category of *ShapeNet* (Chang et al., 2015) and includes diverse vehicle shapes. The fluid dynamics simulations operate with a fixed inlet velocity of $20\,\mathrm{m/s}$ ($72\,\mathrm{km/h}$) and a Reynolds number of $5 \times 10^6$ (Umetani & Bickel, 2018). Each sample discretizes the car surface into approximately 3,700 mesh points. We partition 600 watertight instances into 450 samples for training, 50 samples for validation, and the last 100 samples for testing. The model targets the estimation of scalar *pressure* field across the surface mesh.

**DeepJEB.** To verify the model's applicability to diverse physical systems, we include a solid mechanics benchmark beyond fluid dynamics. The *DeepJEB* dataset (Hong et al., 2025) serves as a standard benchmark for analyzing the structural response of jet engine brackets. The primary objective is the prediction of the vector-valued *deflection* field and the scalar *von Mises stress*. The dataset is simulated under linear static load cases (vertical, horizontal, diagonal, and torsional). Regarding dataset organization, we allocate a random 10% of training samples for validation. The dataset comprises 1,776 samples for training, 162 for validation, and 197 for testing.

The structural responses in *DeepJEB* are computed under the assumption of 3D linear static isotropic elasticity with small-strain kinematics. The dataset labels represent solutions to the equilibrium equation (Landau et al., 2012):

$$\nabla \cdot \boldsymbol{\sigma}(\mathbf{u}) = \mathbf{0}, \tag{13}$$

where $\mathbf{u}$ denotes the displacement field and $\boldsymbol{\sigma}$ is the stress tensor related to strain via the linear isotropic constitutive law

parameterized by Young's modulus $E = 113.8\,\text{GPa}$ and Poisson's ratio $\nu = 0.342$ for the Ti–6Al–4V material (Hong et al., 2025). This system is solved via the finite element method under linear static load cases.

## C. Evaluation Metrics

To evaluate the predictive accuracy of our proposed model and the baseline methods, we employ the *relative $L_2$ error*. Unlike absolute metrics such as the standard $L_1$ or $L_2$ errors, which are sensitive to the original scales of the data, this metric measures the deviation of the predicted physical fields from the ground truth solutions, normalized by the magnitude of the reference. This normalization ensures that the error metric remains scale-invariant, thereby facilitating fair comparisons across diverse samples with varying physical magnitudes.

Formally, let $\mathbf{y}^{(i)} \in \mathbb{R}^{\mathcal{V} \times d}$ denote the ground truth field values (e.g., pressure or velocity) for the $i$-th sample in the dataset, defined on a mesh with $\mathcal{V}$ nodes and $d$ components. Let $\hat{\mathbf{y}}^{(i)}$ denote the corresponding prediction generated by the model. The relative $L_2$ error for a single sample $i$, denoted as $\epsilon_i$, is defined as:

$$\epsilon_i = \frac{\|\hat{\mathbf{y}}^{(i)} - \mathbf{y}^{(i)}\|_F}{\|\mathbf{y}^{(i)}\|_F}, \tag{14}$$

where $\| \cdot \|_F$ represents the Frobenius norm (equivalent to the Euclidean norm of the flattened vector) over the spatial domain.

For the final evaluation on the test set containing $S$ samples, we report the mean relative $L_2$ error averaged over all samples:

$$\mathcal{E}_{\text{test}} = \frac{1}{S} \sum_{i=1}^{S} \epsilon_i. \tag{15}$$

## D. Computational Efficiency and Complexity Analysis

### D.1. Hardware and Software Infrastructure

All experiments were conducted on a workstation running **Rocky Linux 9.3**. The system is equipped with an **Intel Xeon Gold 6530 CPU** and **512GB of RAM**. We utilized **NVIDIA L40S GPUs**, each with **48GB of VRAM**. The software environment was configured with **CUDA 12.4** and PyTorch.

### D.2. Complexity and Latency Comparison

Table 4 presents a comprehensive comparison of model complexity, computational cost, and inference latency. We report the total GFLOPs measured during the forward pass, including all operations such as FFT/IFFT and spectral convolutions.

| Model | Params (M) | Inference (ms) | Training (ms) | GFLOPs |
|---|---|---|---|---|
| PointNet | 3.5 | 3.9 | 4.6 | 1.0 |
| PointNet++ | 1.4 | 139.2 | 112.6 | 2.4 |
| PointDeepONet | 0.2 | 3.1 | 3.8 | 1.5 |
| EGNN | 0.3 | 2.9 | 4.9 | 8.2 |
| EMNN | 0.7 | 10.7 | 10.6 | 15.6 |
| TEMNN | 0.8 | 6.6 | 5.8 | 13.5 |
| MeshGraphNet | 0.4 | 3.6 | 4.4 | 10.5 |
| Transolver | 1.5 | 9.1 | 14.6 | 11.1 |
| GINO | 285.0 | 63.7 | 64.6 | 153.1 |
| **EqGINO** | **4.1** | **52.6** | **57.7** | **85.7** |

*Table 4.* Comparison of computational complexity and latency.

**Discussion.** As shown in Table 4, our proposed EqGINO significantly reduces model size compared to the spectral baseline, GINO. The parameter count is reduced by approximately 98% ($285M \rightarrow 4.1M$) primarily due to the proposed *Orbit-based Weight Sharing* mechanism. Regarding computational cost, EqGINO reduces the total GFLOPs by roughly 44% ($153.1 \rightarrow 85.7$ GFLOPs), a gain attributed to combined factors such as the exclusion of explicit coordinate features and the utilization of channel grouping.

We acknowledge that EqGINO entails higher computational costs compared to lightweight baselines, such as Transolver. This is primarily attributed to the graph construction phase within the GNO modules. Even with hash grid-based optimizations, the neighbor search process scales with a computational complexity of $O(Ndr^3)$ (Li et al., 2023)(where $N, d, r$ denote the number of points, density, and radius, respectively). While manageable for standard benchmarks, this cubic dependency becomes a critical bottleneck for *million-scale datasets*, where the computational burden and memory usage for neighbor queries become prohibitive.

However, a critical advantage of EqGINO is its ability to embed equivariance directly into the spectral domain without an increase in computational complexity compared to the GINO baseline. This intrinsic design proves superior to extrinsic strategies; as demonstrated in Table 1b and 2, even when baselines are trained with extensive data augmentation, they fail to match the robust equivariant performance of our method. While EqGINO successfully incorporates SE(3)-equivariance into the spectral framework, computational cost reduction for million-scale datasets remains a promising direction for future work.

## E. Equivariance for EqGNO

In this section, we provide a proof of equivariance for the EqGNO encoder. The proof for the decoder follows an analogous logic.

Recall that we parameterize the (learnable) kernel function in the encoder solely by relative geometric quantities. Under uniform density, let $\bar{y}$ denote the centroid of the input point cloud $\mathcal{P}$. We use the relative distance $\|x - y_j\|$ and the centroid-relative distance $\|x - \bar{y}\|$. This formulation guarantees that the kernel remains invariant under any rigid transformation $g \in SE(3)$ (where $g \cdot x = Rx + t$):

$$\kappa(g \cdot x, g \cdot y_j; g \cdot \bar{y}) = \kappa(x, y_j; \bar{y}). \tag{16}$$

Note that the centroid shifts consistently with the transformation ($g \cdot \bar{y} = R\bar{y} + t$), which preserves the relative distances.

**Proof of SE(3)-Equivariance.** We now verify that the encoded field preserves the SE(3) symmetry of the input. Let $\mathcal{E}$ denote the encoder that constructs a latent feature field $v_0$ and $\mathcal{P}' = g \cdot \mathcal{P} = \{y'_j\}_{j=1}^{M}$ be the transformed point cloud. The encoded field at a location $x$ becomes

$$\begin{aligned}
[\mathcal{E}(\mathcal{P}')](x) &\approx \sum_j \kappa(x, y'_j; \bar{y}')\mu_j \\
&= \sum_j \kappa(x, g \cdot y_j; g \cdot \bar{y})\mu_j.
\end{aligned} \tag{17}$$

By the SE(3)-invariance property of the kernel ($\kappa(x, g \cdot y) = \kappa(g^{-1} \cdot x, y)$), we obtain

$$\begin{aligned}
[\mathcal{E}(\mathcal{P}')](x) &\approx \sum_j \kappa(g^{-1} \cdot x, y_j; \bar{y})\mu_j \\
&\approx v_0(g^{-1} \cdot x) \\
&= [T_g^* v_0](x).
\end{aligned} \tag{18}$$

This confirms that encoding the transformed point cloud results in the pullback of the original encoded field. Thus, the output of the encoder becomes a valid SE(3)-equivariant feature field, which yields a geometrically consistent initialization for the subsequent spectral layers.

## F. Proof of Lemma 4.1 (Spectral Rotation Property)

**Lemma 4.1** Let $f \in L^2(\mathbb{R}^3)$ be a scalar field and $\mathcal{T}_R$ be the rotation operator defined by $[\mathcal{T}_R f](x) = f(R^{-1}x)$. The Fourier transform satisfies

$$\widehat{\mathcal{T}_R f}(\mathbf{k}) = \hat{f}(R^{-1}\mathbf{k}). \tag{19}$$

*Proof.* The Fourier transform of $[\mathcal{T}_R f](x)$ yields

$$
\begin{aligned}
\widehat{\mathcal{T}_R f}(\mathbf{k}) &= \int_{\mathbb{R}^3} \mathcal{T}_R(x) e^{-2i\pi\langle \mathbf{k}, x \rangle} dx \\
&= \int_{\mathbb{R}^3} f(R^{-1}x) e^{-2i\pi\langle \mathbf{k}, x \rangle} dx
\end{aligned}
\tag{20}
$$

The change of variables $y = R^{-1}x$ implies $dx = dy$ (since $\det R = 1$) and $\langle \mathbf{k}, x \rangle = \langle \mathbf{k}, Ry \rangle = \langle R^{-1}\mathbf{k}, y \rangle$. Consequently,

$$
\begin{aligned}
\widehat{\mathcal{T}_R f}(\mathbf{k}) &= \int_{\mathbb{R}^3} f(y) e^{-2i\pi\langle R^{-1}\mathbf{k}, y \rangle} dy \\
&= \hat{f}(R^{-1}\mathbf{k}).
\end{aligned}
\tag{21}
$$

Thus, a rotation in the spatial domain corresponds to an equivalent rotation of the frequency modes $\mathbf{k}$.

## G. Proof of Isotropic Spectral Convolution

Recall Eq. (10). For any feature field $v_{in} \in \mathcal{H}_{in}$, the equivariant spectral convolution layer satisfies

$$\mathcal{F}\left(\mathcal{L}(T_g^* v_{in})\right)(\mathbf{k}) = \mathcal{F}\left(T_g^*(\mathcal{L}v_{in})\right)(\mathbf{k}). \tag{22}$$

Consider a rigid transformation $g = (R, \mathbf{t}) \in SE(3)$. In the spectral domain, the spatial translation $\mathbf{t}$ manifests as a scalar phase shift $e^{-2\pi i \langle \mathbf{k}, \mathbf{t} \rangle}$, whereas the rotation $R$ corresponds to fetching the value from $R^{-1}\mathbf{k}$. Since the spectral convolution is a linear operation, the scalar phase term appears identically on both sides of the equation and cancels out. Consequently, we limit our derivation to the rotational component $R$ without loss of generality. We expand both sides using the property $\widehat{\mathcal{L}v_{in}}(\mathbf{k}) = W(\mathbf{k})\widehat{v_{in}}(\mathbf{k})$ and the group action defined via the pullback operator (Eq. (2)). For simplicity, we denote $\widehat{v_{out}}(\mathbf{k}) = \widehat{\mathcal{L}v_{in}}(\mathbf{k})$ as the output feature representation.

- **LHS (Spectral Convolution on transformed input):** First, the rotated input $T_g^* v_{in}$ yields Fourier coefficients $\rho_{in}(R^{-1})\widehat{v_{in}}(R^{-1}\mathbf{k})$. The layer applies $W(\mathbf{k})$ to this:

$$\text{LHS} = W(\mathbf{k}) \cdot \left[\rho_{in}(R^{-1})\widehat{v_{in}}(R^{-1}\mathbf{k})\right]. \tag{23}$$

- **RHS (Transformed output of spectral convolution):** The right-hand side represents the transformed output $T_g^* v_{out}$. By Lemma 4.1 with the output representation $\rho_{out}$, we have

$$\text{RHS} = \rho_{out}(R^{-1}) \cdot \widehat{v_{out}}(R^{-1}\mathbf{k}) = \rho_{out}(R^{-1}) \cdot \left[W(R^{-1}\mathbf{k})\widehat{v_{in}}(R^{-1}\mathbf{k})\right]. \tag{24}$$

Equating LHS and RHS for an arbitrary input component $\widehat{v_{in}}(R^{-1}\mathbf{k})$, we obtain

$$W(\mathbf{k})\rho_{in}(R^{-1}) = \rho_{out}(R^{-1})W(R^{-1}\mathbf{k}). \tag{25}$$

To express this constraint for the weight at a rotated frequency, let $\mathbf{q} = R^{-1}\mathbf{k}$ (which implies $\mathbf{k} = R\mathbf{q}$). Substituting $\mathbf{k}$ with $R\mathbf{q}$ yields

$$W(R\mathbf{q})\rho_{in}(R^{-1}) = \rho_{out}(R^{-1})W(\mathbf{q}). \tag{26}$$

Finally, right-multiplying by $\rho_{in}(R^{-1})^{-1} = \rho_{in}(R)$, we derive the *general weight conjugation constraint*:

$$W(R\mathbf{k}) = \rho_{out}(R^{-1})W(\mathbf{k})\rho_{in}(R). \tag{27}$$

***Remark A.1. Group Representations and Properties*** A *group representation* $\rho$ of $\mathcal{G}$ on a vector space $V = \mathbb{R}^d$ is a group homomorphism $\rho : \mathcal{G} \to GL(V)$, where $GL(V)$ denotes the general linear group of invertible $d \times d$ matrices. Being a homomorphism implies two key properties:

1. Identity Preservation: $\rho(I_3) = I_d$.

2. Multiplicativity: $\rho(R_1 R_2) = \rho(R_1)\rho(R_2)$ for all $R_1, R_2 \in SO(3)$.

From the definition, $\rho(R)\rho(R^{-1}) = \rho(RR^{-1}) = \rho(I_3) = I_d$. Thus, $\rho(R^{-1})$ is indeed the unique matrix inverse of $\rho(R)$.

**Remark A.2. Output Representation for Vector Fields** In the case of vector field prediction (e.g., displacement), the output dimension $d_{out}$ transforms non-trivially. If $d_{out}$ consists of $m$ numbers of 3D vectors, the representation $\rho_{out}(R^{-1})$ is the direct sum of $m$ rotation matrices $R^T$:

$$\rho_{out}(R^{-1}) = \underbrace{R^T \oplus R^T \oplus \cdots \oplus R^T}_{m \text{ times}} = \begin{bmatrix} R^T & 0 & \cdots & 0 \\ 0 & R^T & \cdots & 0 \\ \vdots & \vdots & \ddots & \vdots \\ 0 & 0 & \cdots & R^T \end{bmatrix} \in \mathbb{R}^{d_{out} \times d_{out}} \tag{28}$$

The transpose $R^T$ ensures that vector fields transform contravariantly under rotations, thereby preserving rotation equivariance of the prediction (Weiler et al., 2018).

**Remark A.3. Challenges in vector field prediction and the need for a Local Basis** As mentioned in Section 4.3, we only use scalar feature fields which yields $\rho_{in}(R) = I_{d_{in}}$. Ideally, an equivariant operator with vector-valued outputs should satisfy the conjugation constraint $W(R\mathbf{k}) = \rho_{out}(R^{-1})W(\mathbf{k})$. However, in practical engineering simulations, the input geometry often lacks a predefined canonical pose. Without a canonical reference, the rotation matrix $R$ is unknown, making it impossible to determine the inverse rotation $R^{-1}$ required to enforce weight constraints. Consequently, the model cannot distinguish between the intrinsic geometry of the object and its arbitrary orientation in the global coordinate system.

For this reason, we impose the isotropy condition on the learnable weight matrix only when predicting scalar physical quantities. In contrast, predicting vector-valued physical fields poses an additional challenge due to the absence of a known canonical pose. To address this issue, the construction of a local basis is essential, as described in Section 4.4.

# H. Incompatibility of RFFT with Rotation Equivariance

In this section, we rigorously demonstrate that the Real-FFT (RFFT) format in 3D cannot support rotation-equivariant operations due to the violation of complex linearity.

Let $f : \mathbb{R}^3 \to \mathbb{R}$ be a real-valued input signal. Its Discrete Fourier Transform $\hat{f}$ satisfies the *Hermitian symmetry* property:

$$\hat{f}(-\mathbf{k}) = \overline{\hat{f}(\mathbf{k})}, \quad \forall \mathbf{k} \in \mathbb{Z}^3. \tag{29}$$

To improve memory efficiency, RFFT stores Fourier coefficients only for the half-space where the last dimension index is non-negative. We define the *stored domain* $\mathcal{D}_R$ as

$$\mathcal{D}_R = \left\{ (k_x, k_y, k_z) \in \mathbb{Z}^3 \;\middle|\; -\frac{N}{2} \le k_x, k_y < \frac{N}{2}, \; 0 \le k_z \le \frac{N}{2} \right\}. \tag{30}$$

For any frequency $\mathbf{k} \notin \mathcal{D}_R$, the value is implicitly reconstructed via conjugation:

$$\hat{f}(\mathbf{k}) := \overline{\hat{f}(-\mathbf{k})}. \tag{31}$$

A standard neural operator layer in the frequency domain operates as a complex linear transformation $\hat{g}(\mathbf{k}) = W(\mathbf{k})\hat{f}(\mathbf{k})$, where $W(\mathbf{k}) \in \mathbb{C}^{C \times C}$. For rotation equivariance to hold, the rotation operation itself must be representable as a linear map on the feature space. However, Eq. (31) forces the operation to output the conjugate $\overline{\hat{f}(-\mathbf{k})}$. Complex conjugation operation $\mathcal{C}$ which is defined by $\mathcal{C}(z) = \bar{z}$ is anti-linear (or conjugate-linear) operation. Specifically, for a scalar $\alpha \in \mathbb{C}$ with $\text{Im}(\alpha) \neq 0$ and a variable $z$,

$$\mathcal{C}(\alpha z) = \bar{\alpha}\bar{z} = \bar{\alpha}\mathcal{C}(z). \tag{32}$$

Since the RFFT-induced rotation forces a conjugation for certain modes, the transformation cannot be represented by any complex linear matrix multiplication $W(\mathbf{k})$. Therefore, the RFFT format is structurally incompatible with learning rotation-equivariant representations using complex-linear layers.

As established in the 3D case, the fundamental issue stems from the RFFT storage mechanism, which truncates the spectral domain along a specific axis. For any dimension $d \geq 2$, there exists some rotation that maps a coordinate from the stored half-space (of Fourier modes) to the unstored half-space, necessitating conjugate retrieval. Since the resulting anti-linear operation breaks the complex-linearity assumption of the neural operator, we conclude that Full-FFT is strictly required to preserve exact rotation equivariance across any rotation $R \in SO(3)$.

## I. Local Basis Construction and Proof of SE(3)-Equivariance

Local Basis strategy enables the prediction of invariant projection coefficients with respect to rigid transformation. We construct a local reference frame $\{\mathbf{e}_{i,1}, \mathbf{e}_{i,2}, \mathbf{e}_{i,3}\}$ at each point $i$, derived intrinsically from the input physical features. Decompose the target field $\mathbf{v}$ into

$$\mathbf{v}_i = c_{i,1}\mathbf{e}_{i,1} + c_{i,2}\mathbf{e}_{i,2} + c_{i,3}\mathbf{e}_{i,3}. \tag{33}$$

Since both the basis vectors $\mathbf{e}_{i,k}$ and the vector-valued field $\mathbf{v}_i$ rotate simultaneously with the object, the projection coefficients $c_{i,k} = \mathbf{v}_i \cdot \mathbf{e}_{i,k}$ remain invariant under rigid body transformations. A linear combination of these predicted coefficients with the transformed local bases recovers the final equivariant vector field.

**Construction of Local Basis**   For the construction of a local basis, we leverage the global external force vector for the *DeepJEB* (Hong et al., 2025) and the inlet velocity vector for the *AhmedBody* (Ahmed et al., 1984) and *ShapeNetCar* (Umetani & Bickel, 2018; Chang et al., 2015) datasets. For simplicity, denote such an external physical vector as $\mathbf{F}$. Let $\mathbf{x}_i$ be the position of the $i$-th node and $\mathbf{c} = \frac{1}{N} \sum \mathbf{x}_j$ be the centroid of the point cloud. Note that the relative position vector $\mathbf{r}_i := \mathbf{x}_i - \mathbf{c}$ is translation-invariant. The orthonormal basis vectors $\{\mathbf{e}_{i,1}, \mathbf{e}_{i,2}, \mathbf{e}_{i,3}\}$ are constructed as follows:

- **Axis 1:** Align the first axis with a global physical vector $\mathbf{F}$, as it provides the primary change in physical quantity.

$$\mathbf{e}_{i,1} = \frac{\mathbf{F}}{\|\mathbf{F}\|} \tag{34}$$

- **Axis 2:** To capture the local orientation, we use the cross product of the relative position and the primary vector.

$$\mathbf{e}_{i,2} = \frac{\mathbf{r}_i \times \mathbf{e}_{i,1}}{\|\mathbf{r}_i \times \mathbf{e}_{i,1}\|} \tag{35}$$

- **Axis 3:** The basis is completed by the right-hand rule.

$$\mathbf{e}_{i,3} = \mathbf{e}_{i,1} \times \mathbf{e}_{i,2} \tag{36}$$

The resulting local basis matrix is $B_i = [\mathbf{e}_{i,1}, \mathbf{e}_{i,2}, \mathbf{e}_{i,3}]$.

**Proof of Local Basis SE(3)-Equivariance**   Let $g = (R, \mathbf{t}) \in SE(3)$ be a rigid body transformation defined by a rotation matrix $R \in SO(3)$ and a translation vector $\mathbf{t} \in \mathbb{R}^3$. The transformed inputs are

$$\mathbf{x}'_i = R\mathbf{x}_i + \mathbf{t}, \quad \mathbf{F}' = R\mathbf{F}. \tag{37}$$

Note that $\mathbf{F}$ is a vector quantity and thus invariant to translation. The related position vector $\mathbf{r}_i$ is translation-invariant and rotates with the object

$$\mathbf{r}'_i = \mathbf{x}'_i - \mathbf{c}' = (R\mathbf{x}_i + \mathbf{t}) - (R\mathbf{c} + \mathbf{t}) = R(\mathbf{x}_i - \mathbf{c}) = R\mathbf{r}_i. \tag{38}$$

Derive the transformation of the axes using the property that rotation matrices preserve norms (isometry) and distribute over the cross product (i.e., $(R\mathbf{a}) \times (R\mathbf{b}) = R(\mathbf{a} \times \mathbf{b})$):

- **Axis 1:**

$$\mathbf{e}'_{i,1} = \frac{\mathbf{F}'}{\|\mathbf{F}'\|} = \frac{R\mathbf{F}}{\|R\mathbf{F}\|} = R\frac{\mathbf{F}}{\|\mathbf{F}\|} = R\mathbf{e}_{i,1} \tag{39}$$

- **Axis 2:**

$$\mathbf{e}'_{i,2} = \frac{\mathbf{r}'_i \times \mathbf{e}'_{i,1}}{\|\mathbf{r}'_i \times \mathbf{e}'_{i,1}\|} = \frac{(R\mathbf{r}_i) \times (R\mathbf{e}_{i,1})}{\|R(\mathbf{r}_i \times \mathbf{e}_{i,1})\|} = \frac{R(\mathbf{r}_i \times \mathbf{e}_{i,1})}{\|\mathbf{r}_i \times \mathbf{e}_{i,1}\|} = R\mathbf{e}_{i,2} \tag{40}$$

- **Axis 3:**

$$\mathbf{e}'_{i,3} = \mathbf{e}'_{i,1} \times \mathbf{e}'_{i,2} = (R\mathbf{e}_{i,1}) \times (R\mathbf{e}_{i,2}) = R(\mathbf{e}_{i,1} \times \mathbf{e}_{i,2}) = R\mathbf{e}_{i,3} \tag{41}$$

Since each basis vector rotates by $R$, the basis matrix $B_i = [\mathbf{e}_{i,1}, \mathbf{e}_{i,2}, \mathbf{e}_{i,3}]$ satisfies

$$B'_i = [R\mathbf{e}_{i,1}, R\mathbf{e}_{i,2}, R\mathbf{e}_{i,3}] = R \cdot B_i. \tag{42}$$

This confirms that the local basis is SE(3)-equivariant.

## J. Discretization-convergence and Ablation Studies

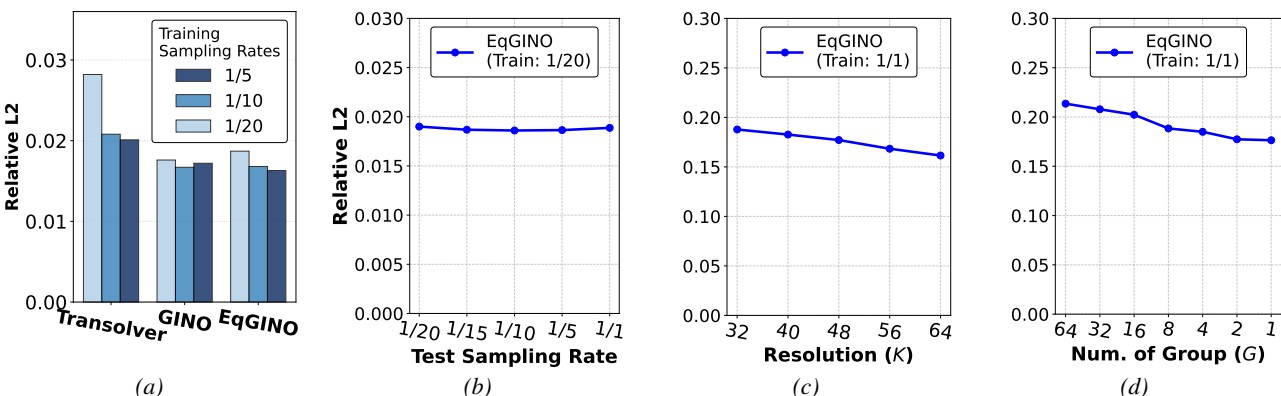

*Figure 7.* Performance evaluation and ablation studies. All models are canonically trained. (a–b) Robustness analysis on *AhmedBody*: (a) impact of varying *training* mesh sampling rates, and (b) generalization to unseen *test* sampling rates with a model trained on sparse (1/20) data. (c–d) Ablation studies on *ShapeNetCar* investigating the effect of (c) spectral resolution per axis, $K$, and (d) number of channel groups, $G$.

In this section, we provide a comprehensive evaluation of the discretization-convergence properties and ablation studies for EqGINO. All models presented in these experiments were trained exclusively on the canonical trainset.

**Discretization-Convergence and Robustness.** EqGINO is designed to effectively embed equivariance while inheriting the discretization-convergence properties inherent to the GINO architecture. This design allows the model to learn resolution-independent operators rather than overfitting to specific mesh discretizations.

As illustrated in Fig. 7a, we evaluated models trained on datasets with varying mesh sampling rates $(1/5, 1/10, 1/20)$ and tested them on the original high-resolution data. While baselines like Transolver degrade in performance as the training data becomes sparser, both GINO and EqGINO demonstrate robust performance across all sampling rates. It is important to note that unlike GINO, EqGINO does not utilize directional information (e.g., coordinate values) to ensure rotation equivariance, which could theoretically limit its expressivity. However, our results show that EqGINO achieves performance comparable to GINO. This suggests that our architectural design—specifically employing a *block-diagonal structure* to enable higher spectral resolution—effectively compensates for the lack of directional cues, maintaining high expressivity even with sparse training data.

Furthermore, Fig. 7b validates the zero-shot super-resolution capability of our model. An EqGINO model trained only on a coarse $1/20$ subsampled dataset successfully generalizes to unseen test sampling rates, maintaining consistent error rates even when evaluated on the full $1/1$ resolution.

**Ablation Studies on Hyperparameters.** We further investigate the impact of spectral resolution $(K)$ and the number of channel groups $(G)$ on model performance using the *ShapeNetCar* dataset.

Fig. 7c indicates that increasing the spectral resolution per axis ($K$) leads to a monotonic decrease in relative $L_2$ error. Since EqGINO employs strict weight-sharing to satisfy equivariance, the number of unique learnable parameters is drastically reduced. However, as discussed in *Remark 2*, a higher $K$ yields a larger number of distinct radial orbits. This effectively increases the number of independent learnable parameters, thereby recovering the model's expressive capacity within a compact parameter budget.

Fig. 7d analyzes the effect of the number of channel groups ($G$). We observe that performance improves as $G$ decreases. However, a smaller $G$ increases computational cost. To mitigate the overhead of Full-FFT, we enforce a block-diagonal structure on the spectral weight tensor. This constraint decreases the FLOPs of channel mixing by a factor of $G$, effectively offsetting the extra computation required for the full set of Fourier modes. We observe that performance remains relatively robust within the range of $G = 4$ to $8$. Therefore, we select a value from this interval to balance the trade-off between computational efficiency (via reduced channel interaction) and predictive accuracy.

## K. Qualitative Analysis of Vector Field Equivariance

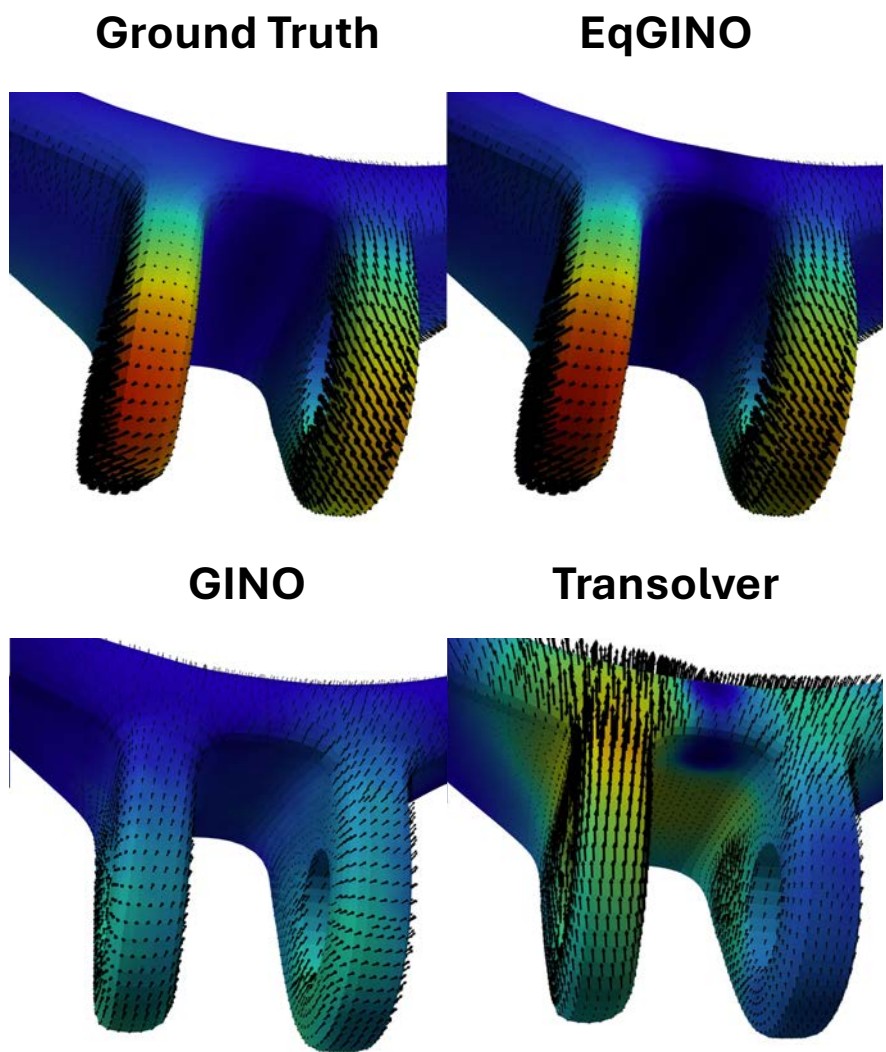

*Figure 8.* Visualization of *deflection* vectors on the *DeepJEB* dataset under torsional loading. Note that all models were trained exclusively on the canonical dataset but evaluated on input geometries rotated by $180°$. The black arrows depict the deflection vectors at each target node, while the contours denote the magnitude of the deflection.

In Figure 8, we qualitatively analyze the vector field predictions. While EqGINO consistently predicts accurate deflection vectors regardless of the input geometry's orientation, state-of-the-art coordinate-dependent models such as GINO and Transolver exhibit significant failures. When these baselines encounter input geometries in unseen positions (i.e., rotated states not present in the training set), they generate completely erroneous prediction patterns that deviate from the ground truth physics. This suggests that the baselines overfit to the absolute spatial coordinates of the training data. Consequently, our results demonstrate that EqGINO is capable of robustly predicting vector fields in an equivariant manner, strictly adhering to the geometric transformation of the input.

## L. Additional Visualizations

### L.1. DeepJEB

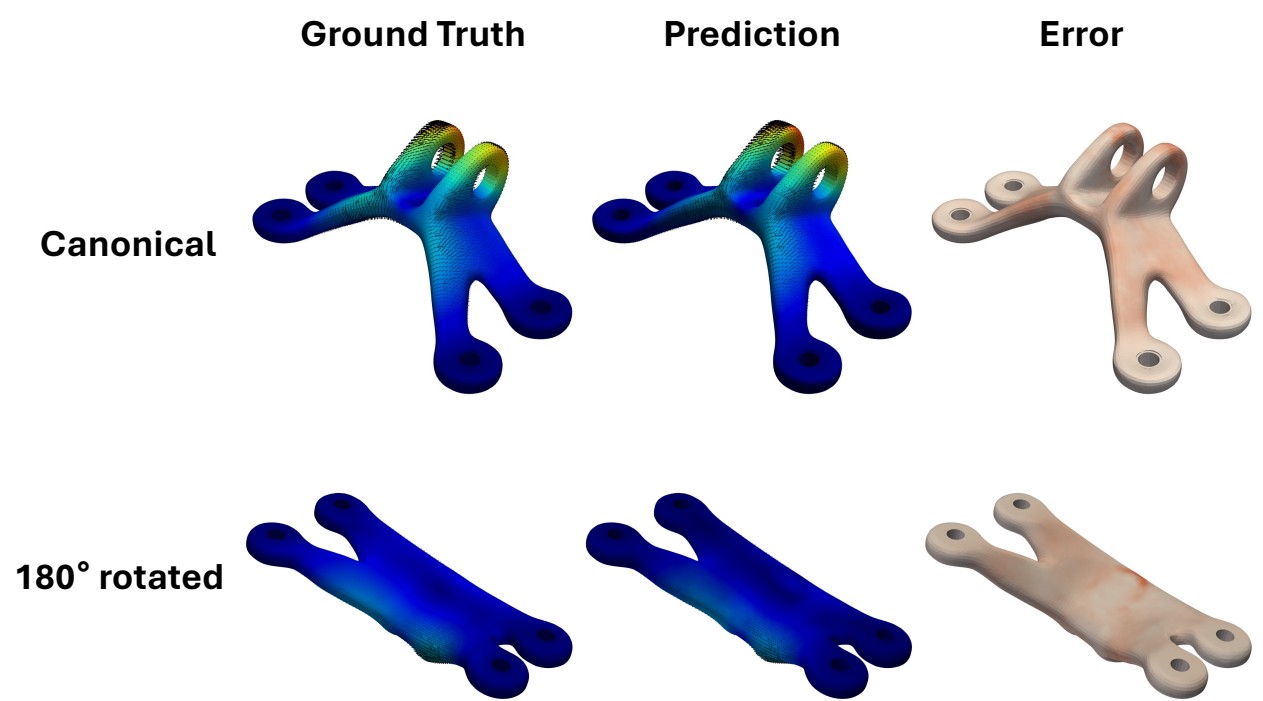

*Figure 9.* Visualization of *deflection* predictions on the *DeepJEB* dataset using **EqGINO**. The model was trained canonically. The rows display the input geometries in different orientations: **(Top)** Canonical input; **(Bottom)** Input rotated by $180°$.

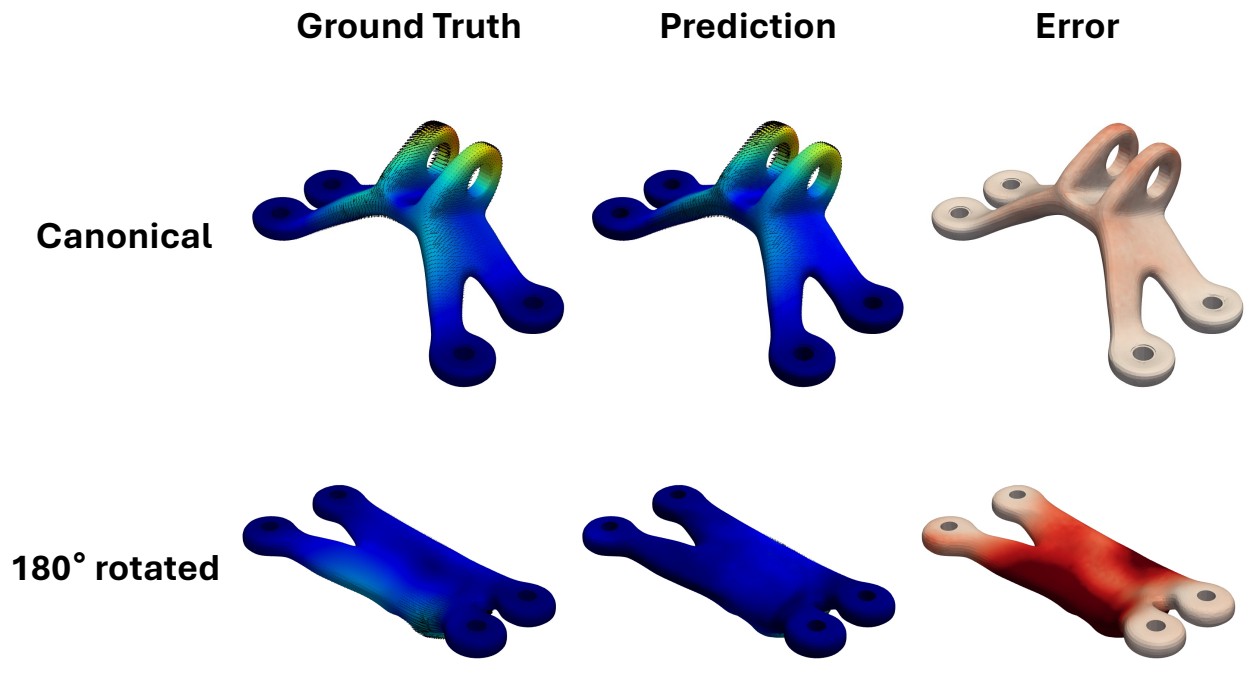

*Figure 10.* Visualization of *deflection* predictions on the *DeepJEB* dataset using **GINO**. The model was trained canonically. The rows display the input geometries in different orientations: **(Top)** Canonical input; **(Bottom)** Input rotated by $180°$.

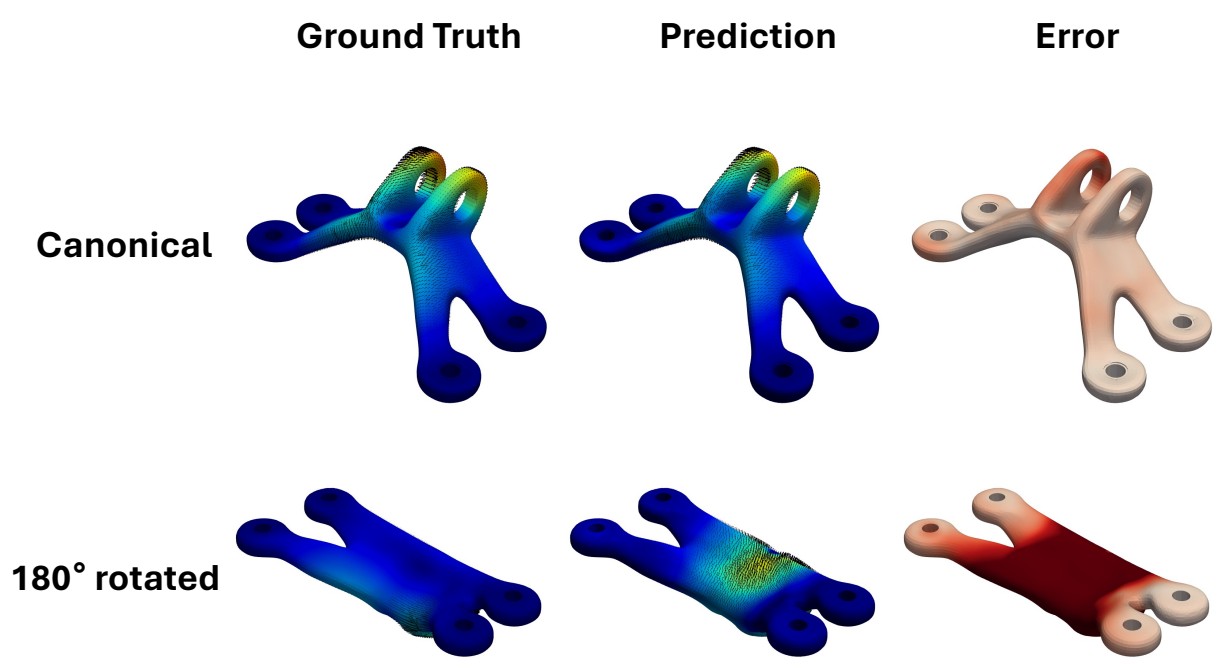

*Figure 11.* Visualization of *deflection* predictions on the *DeepJEB* dataset using **Transolver**. The model was trained canonically. The rows display the input geometries in different orientations: **(Top)** Canonical input; **(Bottom)** Input rotated by $180°$.

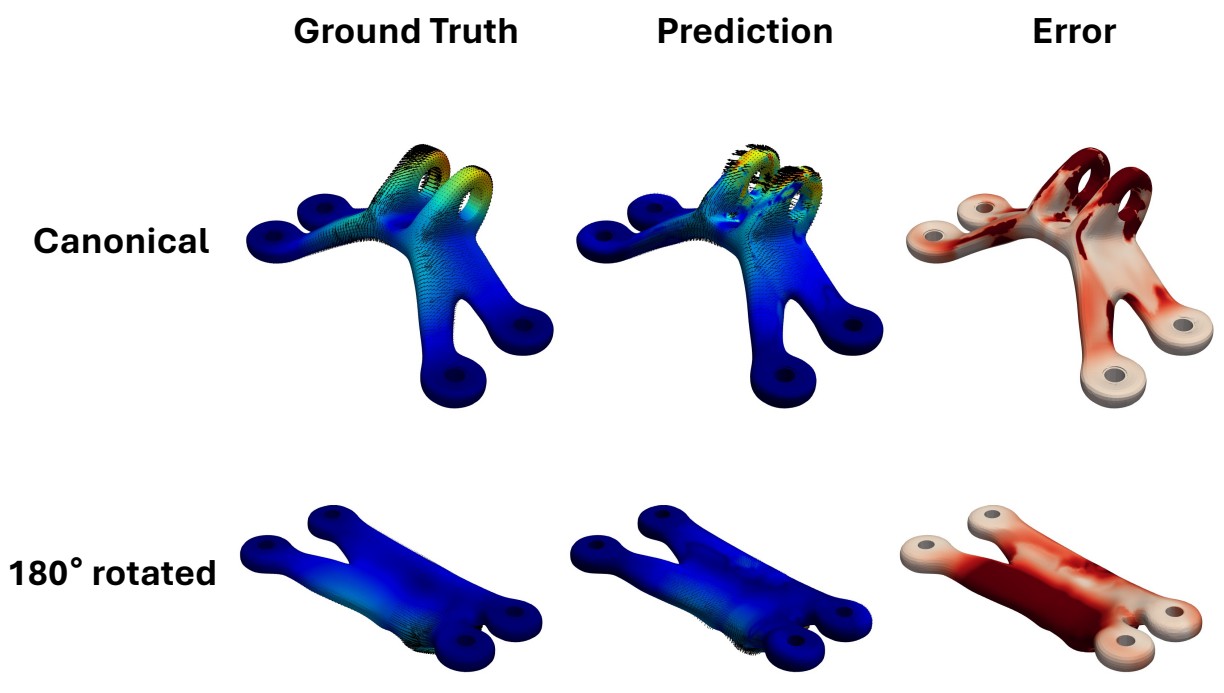

*Figure 12.* Visualization of *deflection* predictions on the *DeepJEB* dataset using **Transolver\***, an SE(3)-equivariant variant of the standard Transolver. The model was trained canonically. The rows display the input geometries in different orientations: **(Top)** Canonical input; **(Bottom)** Input rotated by $180°$.

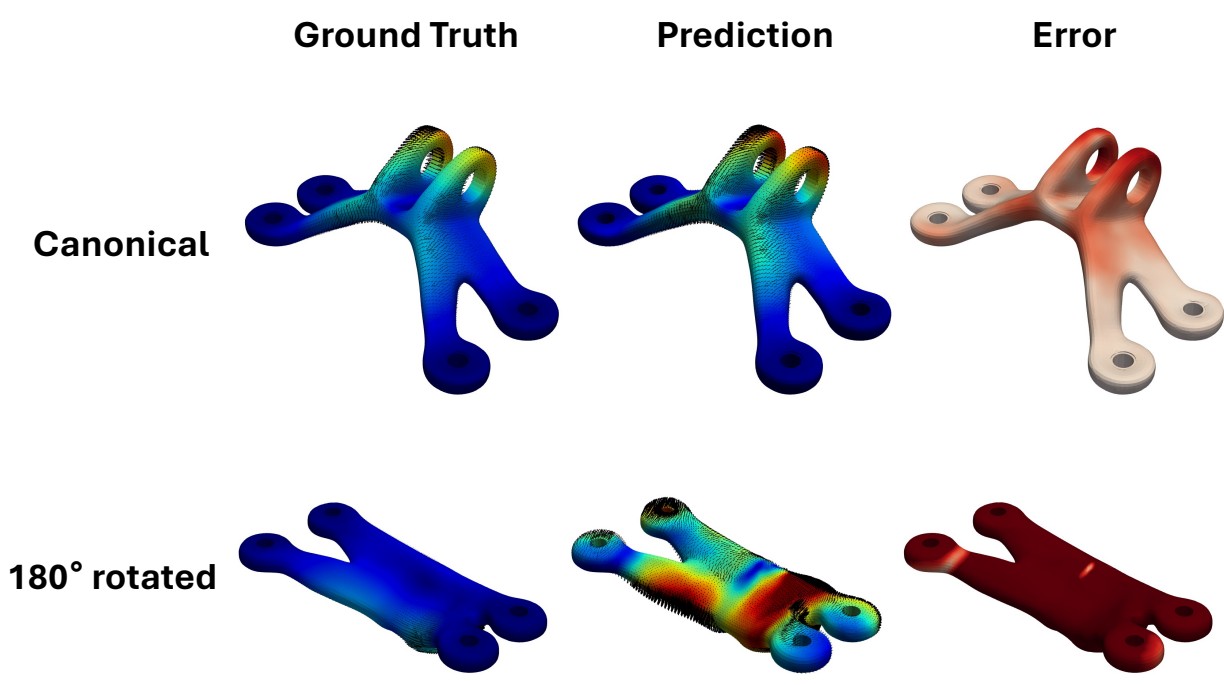

*Figure 13.* Visualization of *deflection* predictions on the *DeepJEB* dataset using **PointNet**. The model was trained canonically. The rows display the input geometries in different orientations: **(Top)** Canonical input; **(Bottom)** Input rotated by $180°$.

## L.2. ShapeNetCar

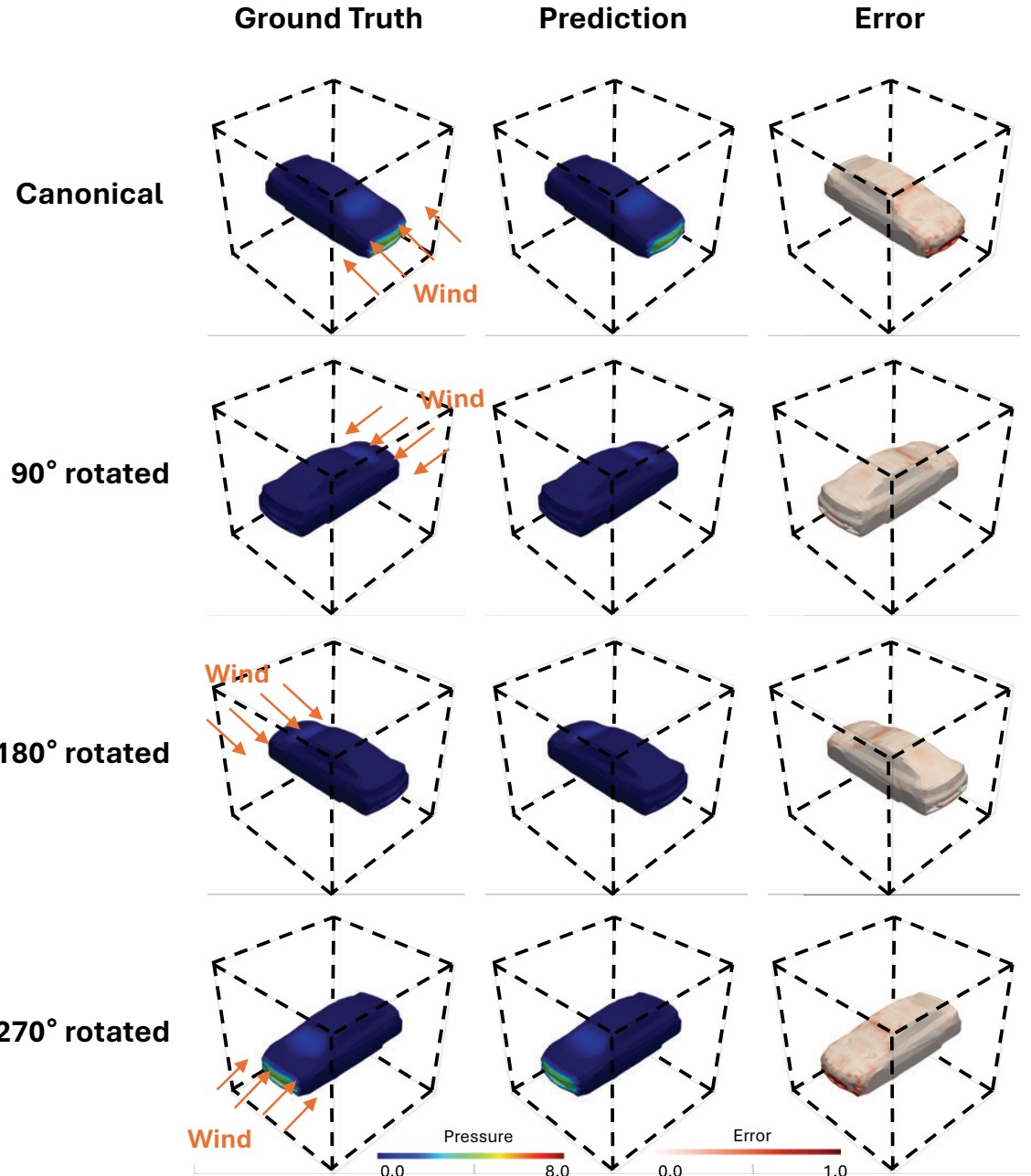

*Figure 14.* Visualization of *pressure* predictions on the *ShapeNetCar* dataset using **EqGINO**. The model was trained canonically. The rows display input geometries in different orientations: **(Top)** Canonical input; **(Rows 2–4)** Inputs rotated by $90°$, $180°$, and $270°$, respectively.

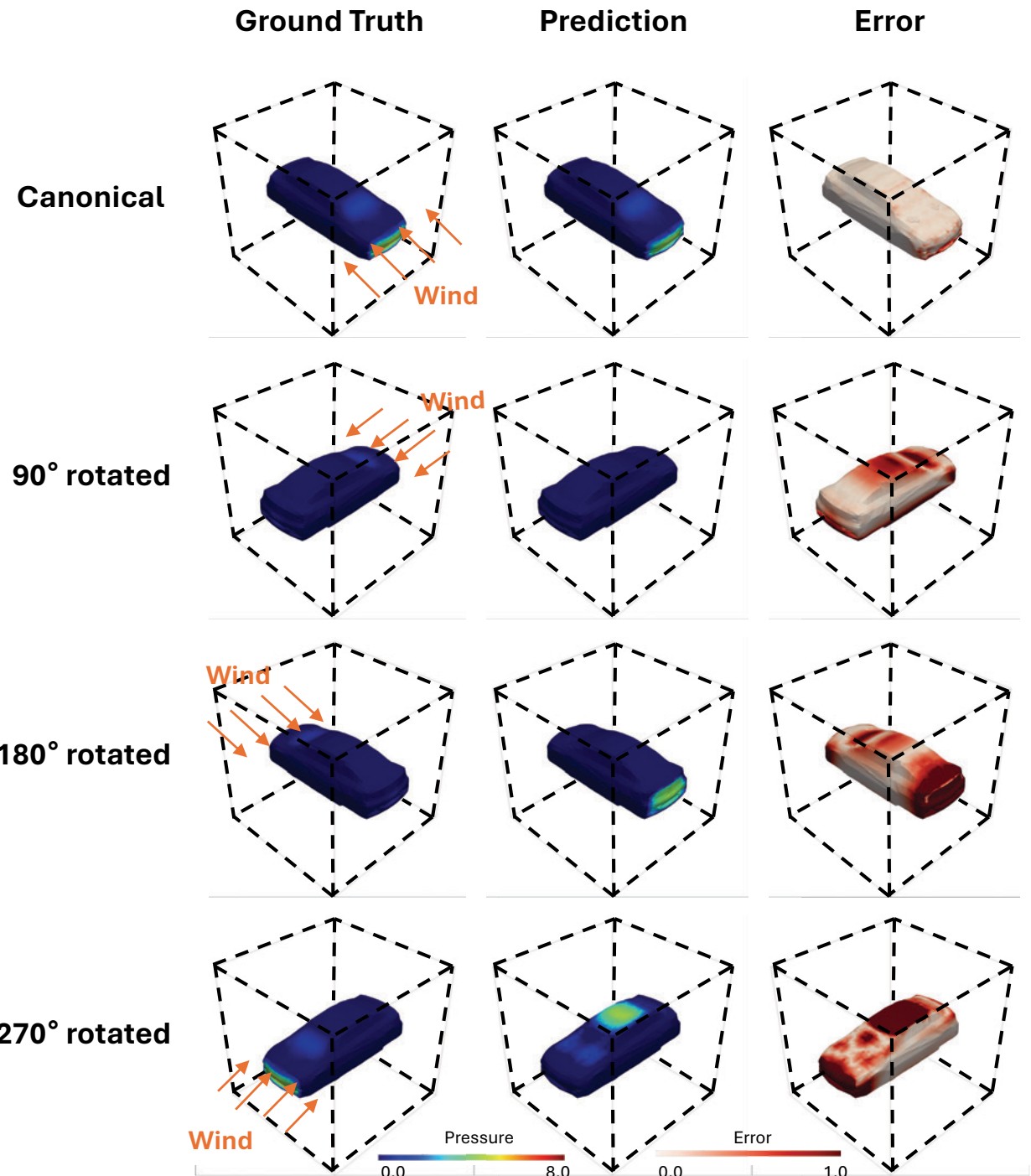

*Figure 15.* Visualization of *pressure* predictions on the *ShapeNetCar* dataset using **GINO**. The model was trained canonically. The rows display input geometries in different orientations: **(Top)** Canonical input; **(Rows 2–4)** Inputs rotated by $90°$, $180°$, and $270°$, respectively.

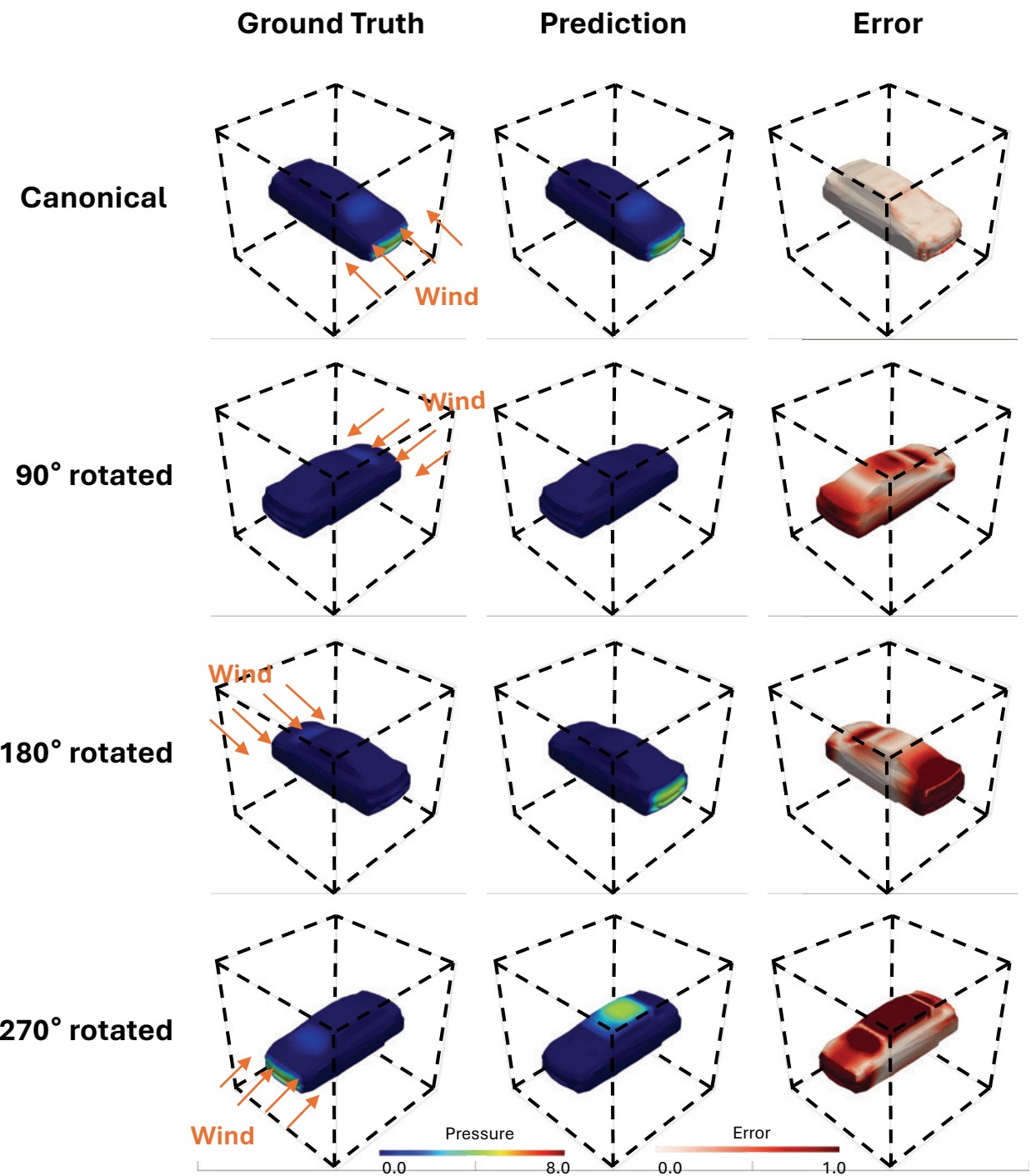

*Figure 16.* Visualization of *pressure* predictions on the *ShapeNetCar* dataset using **Transolver**. The model was trained canonically. The rows display input geometries in different orientations: **(Top)** Canonical input; **(Rows 2–4)** Inputs rotated by $90°$, $180°$, and $270°$, respectively.

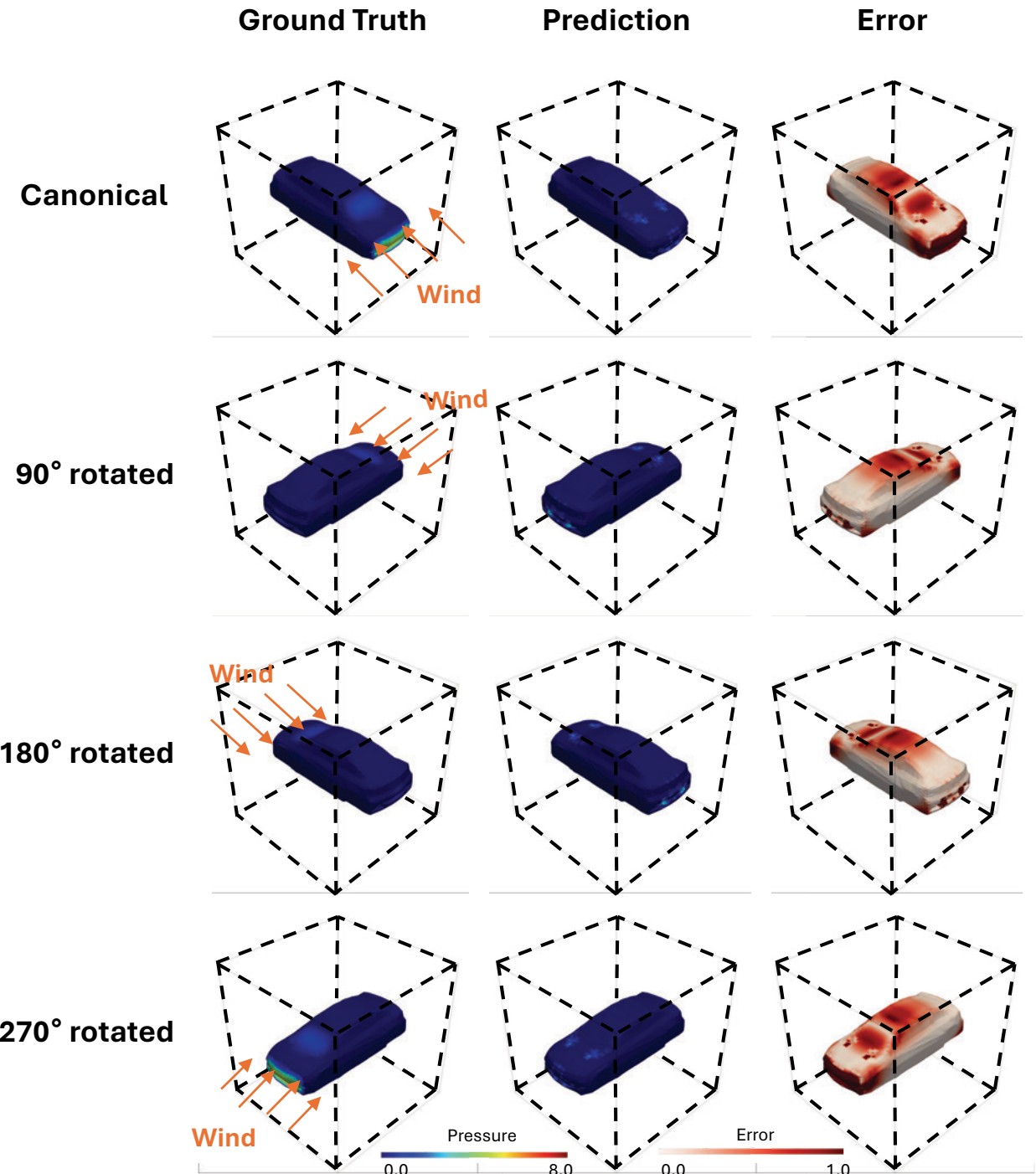

*Figure 17.* Visualization of *pressure* predictions on the *ShapeNetCar* dataset using **Transolver\***, an SE(3)-equivariant variant of the standard Transolver. The model was trained canonically. The rows display input geometries in different orientations: **(Top)** Canonical input; **(Rows 2–4)** Inputs rotated by $90°$, $180°$, and $270°$, respectively.

## L.3. AhmedBody

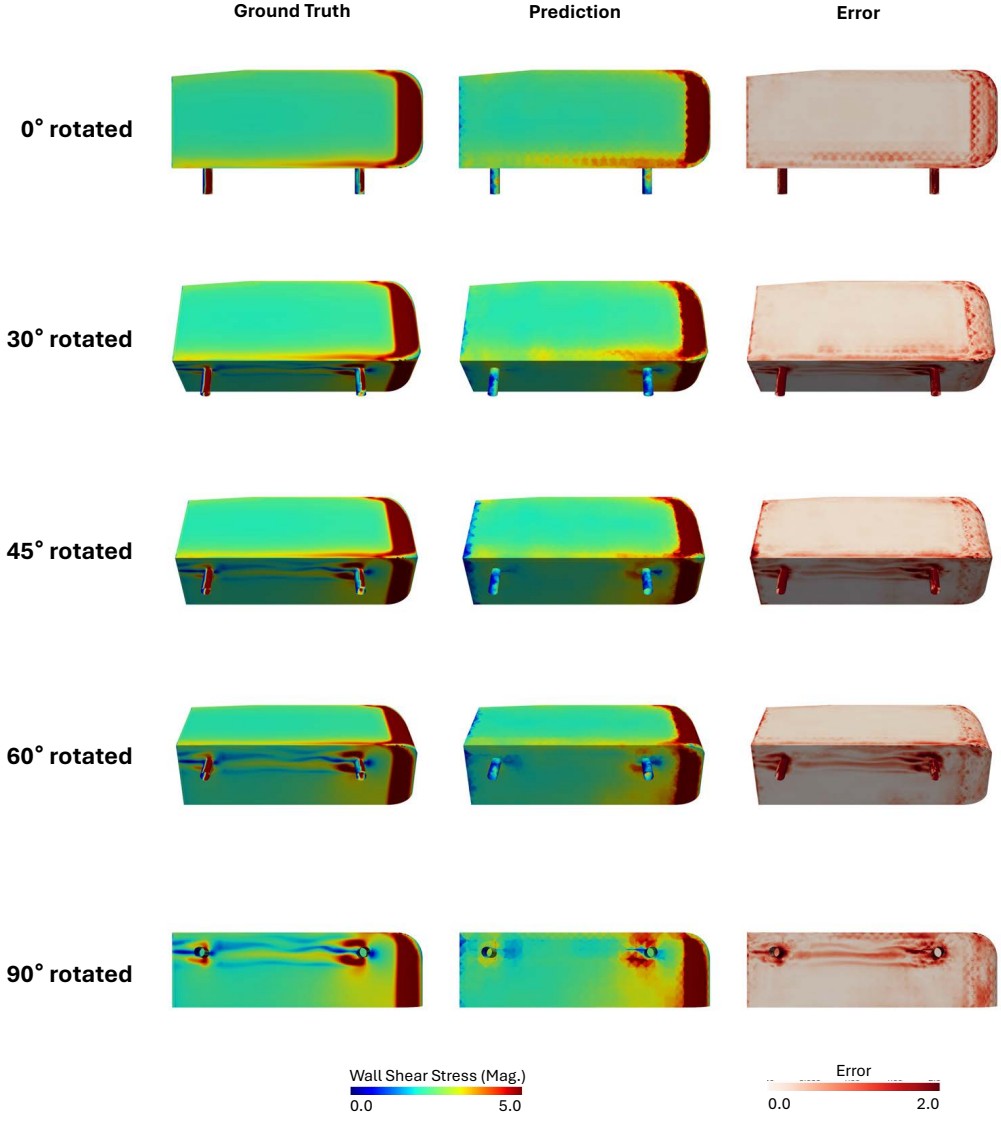

*Figure 18.* Visualization of *wall shear stress* predictions on the *AhmedBody* dataset using **EqGINO**. The model was trained on data with random continuous rotations (i.e., the rotated-to-rotated setting). The rows display input geometries in different orientations: **(Rows 1–5)** Inputs rotated by $0°$, $30°$, $45°$, $60°$, and $90°$, respectively.

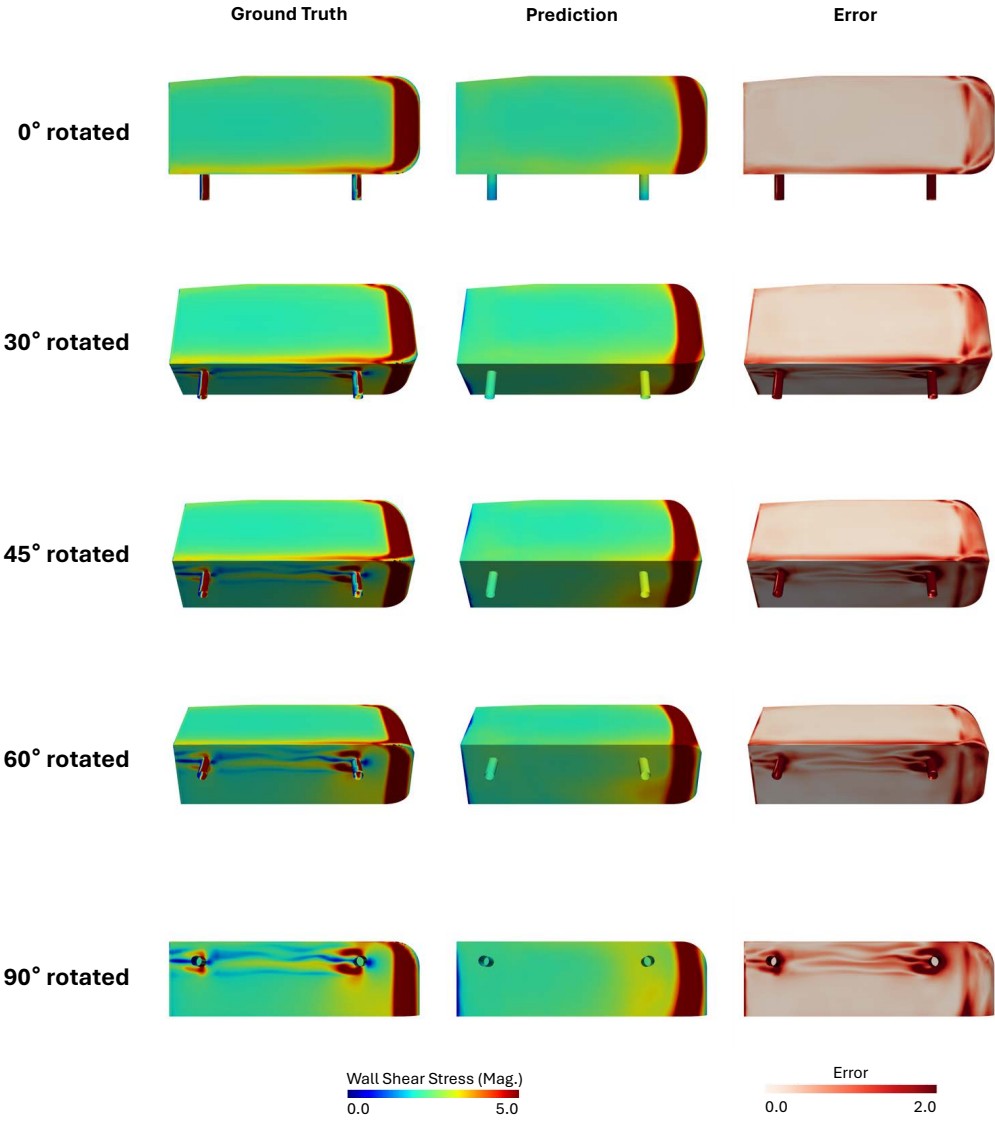

*Figure 19.* Visualization of *wall shear stress* predictions on the *AhmedBody* dataset using **GINO**. The model was trained on data with random continuous rotations (i.e., the rotated-to-rotated setting). The rows display input geometries in different orientations: **(Rows 1–5)** Inputs rotated by $0°$, $30°$, $45°$, $60°$, and $90°$, respectively.

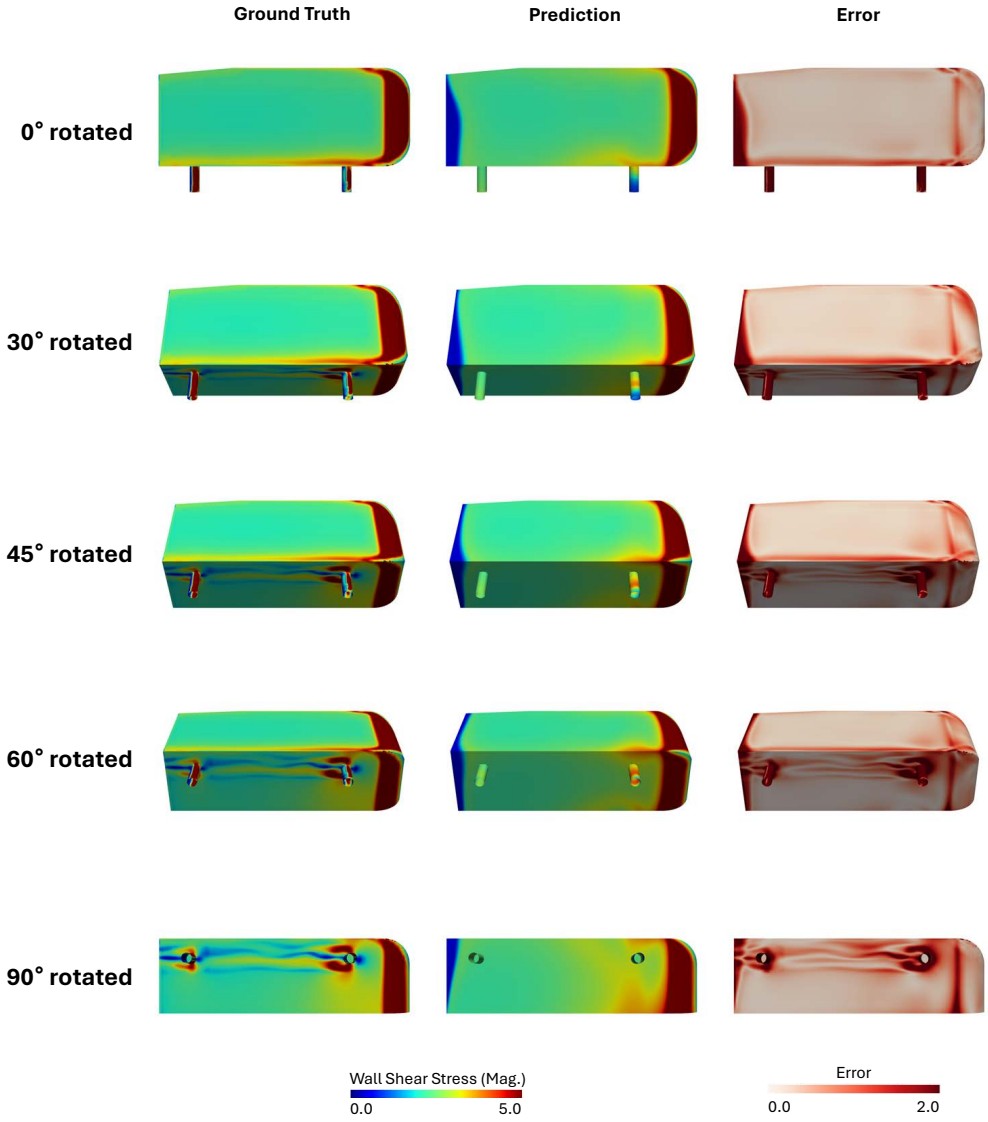

*Figure 20.* Visualization of *wall shear stress* predictions on the *AhmedBody* dataset using **Transolver**. The model was trained on data with random continuous rotations (i.e., the rotated-to-rotated setting). The rows display input geometries in different orientations: **(Rows 1–5)** Inputs rotated by $0°$, $30°$, $45°$, $60°$, and $90°$, respectively.

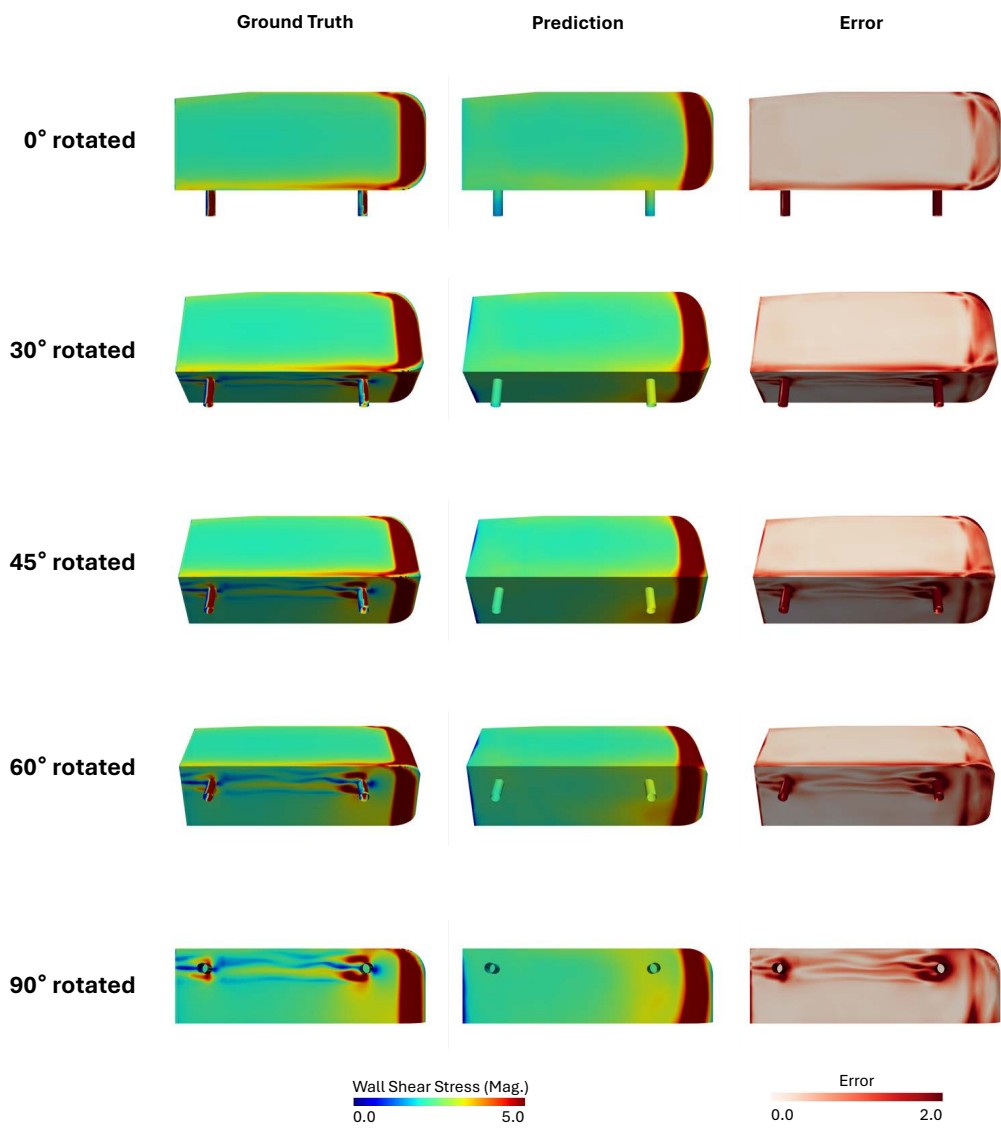

*Figure 21.* Visualization of *wall shear stress* predictions on the *AhmedBody* dataset using **PointNet**. The model was trained on data with random continuous rotations (i.e., the rotated-to-rotated setting). The rows display input geometries in different orientations: (**Rows 1–5**) Inputs rotated by $0°$, $30°$, $45°$, $60°$, and $90°$, respectively.

