# OpenReview forum: "EqGINO: Equivariant Geometry-Informed Fourier Neural Operators for 3D PDEs"
_ICML.cc/2026/Conference — ICML 2026 regular_

### Official Review · Reviewer_35pu · 2026-02-14

**Soundness:** 3
**Presentation:** 3
**Significance:** 3
**Originality:** 3
**Overall Recommendation:** 3
**Confidence:** 5

**Summary:**

This paper introduces **EqGINO**, an equivariant geometry-informed neural operator for 3D PDE surrogate modeling, with the goal of reducing overfitting to coordinate systems and improving generalization under rotations.
The method reformulates a GINO-style architecture into two equivariant building blocks: first, an equivariant geometric module (**EqGNO**) that employs a rotation-invariant kernel parameterization, and second, an equivariant Fourier module (**EqFNO**) that imposes spectral isotropy through orbit-based weight sharing in the frequency domain.
The paper further argues that using Full-FFT rather than RFFT is required in order to preserve rotation equivariance during spectral transformations, and it proposes grouping strategies so that the computational cost remains comparable to that of standard FFT-based baselines.
Experimentally, the method is evaluated on both standard in-distribution settings and more demanding rotation-generalization scenarios—including zero-shot tests under discrete rotations and continuous-rotation evaluations with only a limited number of transformed training samples—across several 3D physics and geometry benchmarks.
The submission also presents qualitative results suggesting that EqGINO depends less heavily on absolute coordinate information than coordinate-sensitive baseline methods.

**Compliance With Llm Reviewing Policy:**

Affirmed.

**Key Questions For Authors:**

1. **When does isotropy hurt?** Can you provide targeted experiments (or analysis) on strongly anisotropic PDE settings to quantify the expressivity tradeoff of isotropic spectral weights?
2. **Continuous-rotation behavior:** How does continuous-rotation error scale with grid resolution and/or the amount of SE(3) augmentation? Do errors diminish as resolution increases?
3. **Orbit construction details:** How are frequency “orbits” defined on the integer lattice in practice (ties/degeneracies), and do approximate orbit groupings affect equivariance or accuracy?
4. **Efficiency accounting:** Can you report end-to-end runtime/memory (training + inference) under matched accuracy targets, clarifying the overhead of Full-FFT and the effect of grouping on the tradeoff?
5. **Reproducibility:** Please replace the placeholder code URL with an actual anonymous link and confirm it includes preprocessing, configs, and scripts to reproduce the main results.

**Limitations:**

(i) Exact equivariance is guaranteed only for discrete grid symmetries, while continuous-rotation behavior is empirical;
(ii) Spectral isotropy may reduce expressivity in anisotropic PDE regimes;
(iii) It is recommended to add a "requirements.txt" file to include all the necessary environment packages;
(iv) Results may depend on grid resolution and geometry complexity;
(v) The font size of the table is too small.

**Strengths And Weaknesses:**

### Strengths
- **Soundness:** The architectural design is well aligned with the goal of enforcing discrete rotation equivariance on grid-based 3D domains: combining isotropic spectral weights with orbit-based parameter sharing fits naturally with discrete symmetry groups, while the EqGNO kernel formulation is consistent with rotation-invariant geometric message passing.
- **Clear motivation:** The paper addresses a genuine challenge in 3D PDE surrogate modeling—undesired sensitivity to coordinate frames—and responds with a method that explicitly builds rotational structure into the model while still preserving the global receptive-field advantages associated with Fourier-based operators.
- **Useful practical insight:** The paper’s observation that Full-FFT, rather than RFFT, is required to preserve equivariance in spectral transformations is a practically valuable takeaway for researchers and practitioners implementing equivariant Fourier models.
- **Efficiency-aware design:** The use of grouping and channel-block strategies provides a sensible mechanism for controlling computation and reducing parameter count without abandoning the equivariance constraints built into the architecture.
- **Evaluation focuses on the right stress test:** The experimental setup directly examines the model under the kinds of conditions most relevant to the paper’s claims, including zero-shot generalization to discrete rotations and continuous-rotation testing with only a small amount of transformed training data.
- **Qualitative sanity checks:** The visual examples are helpful in showing that EqGINO is less prone to the obvious coordinate-dependent artifacts that appear in baseline methods relying more heavily on absolute positional information.

### Weaknesses
- **Expressivity tradeoff from isotropy:** Imposing spectral isotropy may discard useful directional information. Although the paper acknowledges that this choice could reduce expressiveness in principle, the analysis would be more convincing if it more clearly identified the settings in which this becomes harmful, such as anisotropic PDE behavior, boundary-layer effects, or direction-dependent forcing.
- **Discrete vs. continuous equivariance gap:** The method provides exact equivariance only with respect to the discrete symmetries supported by the underlying discretized grid, whereas its behavior under continuous rotations is demonstrated empirically rather than guaranteed theoretically. A more explicit discussion of where this gap matters, and how the corresponding error changes with grid resolution, would improve the paper’s clarity and technical grounding.
- **Comparability of baselines:** Since there is related prior work on group-equivariant neural operators and spectral architectures, the empirical case would be stronger with clearer positioning relative to that literature and with more standardized matching of computational budget and parameter count across baseline methods.

---

> ### Author Rebuttal · Authors · 2026-03-31
>
> We are grateful for the reviewer's careful evaluation and constructive suggestions.
>
> ---
>
> ### Response to W1 and Q1 (35pu):
> We agree that isotropic constraints limit the capacity to capture anisotropic features. The AhmedBody dataset represents the exact challenging conditions you mentioned, featuring a thin boundary layer and direction-dependent forcing. Predicting *wall shear stress* (WS) is heavily dominated by these anisotropic dynamics.
>
> To quantify this tradeoff, Table 1 shows EqGINO achieves competitive or superior WS prediction against non-equivariant baselines. By reducing parameter complexity to $O(K)$, we can substantially increase spectral resolution $K$. This expanded frequency bandwidth compensates for the isotropic restriction, providing enough capacity to resolve fine-grained physical gradients. Additionally, as detailed in Appendix Table 2, we inject directional information via local basis projection, allowing the feature space to process direction-dependent forcing while preserving equivariance.
>
> ---
>
> ### Response to W2 and Q2 (35pu):
> Due to space constraints, we provide a detailed analysis with new experiments and an illustrative figure in our **[Response to Q2 (XePo)](https://openreview.net/forum?id=43QdwsbcNM&noteId=znQBvcGx0L)**, which directly addresses this point. We hope the reviewer understands, and we are happy to discuss further if needed.
>
> ---
>
> ### Response to W3 (35pu):
> We thank the reviewer for this important question. The most closely related prior work is GFNO (Helwig et al., ICML 2023), which achieves group equivariance on 2D regular grids by replicating spectral weights for each element of a discrete group G (e.g., p4 with |G|=4 for 90° rotations, p4m with |G|=8 including reflections), resulting in O(|G| × K²) parameter complexity. Extending this to 3D with the octahedral rotation group (|G|=24) would yield O(24 × K³) spectral parameters. Based on GINO's known resource usage (285M params, 11.7 GB GPU memory at O(K³)), we estimate a 3D GFNO would require ~6.8B parameters and ~255 GB GPU memory—far exceeding any single GPU (A100 80GB). We attempted this extension but encountered out-of-memory errors. EqGINO guarantees equivariance for the same octahedral group at only O(K) parameter complexity (4.1M params, 6.6 GB), a reduction by a factor of 24K². For computational budget matching, we kindly refer to our **[Response to W4 and Q2 (iL68)](https://openreview.net/forum?id=43QdwsbcNM&noteId=7qvOIMdZ4G)**, where we provide detailed parameter-matched experiments with Transolver. We hope the reviewer understands, and we are happy to discuss further if needed.
>
> ---
>
> ### Response to Q3 (35pu):
> The term "orbit" refers to a grouping of frequency modes that must share the same weight to satisfy $W(R\mathbf{k}) = W(\mathbf{k})$. On the integer lattice, two modes $\mathbf{k}_1, \mathbf{k}_2 \in \mathbb{Z}^3$ belong to the same orbit iff $|\mathbf{k}_1|_2 = |\mathbf{k}_2|_2$—what matters is that modes at the same distance from the origin share a single weight.
> In implementation, we assign orbit ID $= k_x^2 + k_y^2 + k_z^2$ and construct a weight dictionary keyed by this ID. Since the ID is computed in integer arithmetic, no floating-point ambiguity or approximate grouping is involved, and equivariance is preserved exactly for the discrete symmetries of the grid.
>
> ---
>
> ### Response to Q4 (35pu):
>
> We provide a detailed ablation on the grouping factor $G$, which controls the trade-off between accuracy, speed, and GPU memory. $G=1$ corresponds to Full-FFT without channel grouping.
>
> | $G$ | s/epoch | Rel $L_2$ | Total Time | Speedup | GPU Mem | Mem Save |
> |:---:|:---:|:---:|:---:|:---:|:---:|:---:|
> | 1 | 182.6s | 0.1764 | 304 min | 1.0× | 7,957 MB | — |
> | 2 | 104.6s | 0.1774 | 174 min | 1.7× | 6,600 MB | ↓17% |
> | 4 | 75.4s | 0.1850 | 126 min | 2.4× | 4,117 MB | ↓48% |
> | 8 | 61.9s | 0.1884 | 103 min | 2.9× | 3,073 MB | ↓61% |
> | 16 | 57.7s | 0.2023 | 96 min | 3.2× | 2,469 MB | ↓69% |
> | 32 | 58.6s | 0.2135 | 98 min | 3.1× | 2,205 MB | ↓72% |
> | 64 | 50.7s | 0.2078 | 85 min | 3.6× | 2,025 MB | ↓75% |
>
> We also measured time-to-accuracy to assess convergence speed. To reach Rel $L_2 \leq 0.21$:
>
> | $G$ | Time to ≤ 0.21 | Speedup |
> |:---:|:---:|:---:|
> | 1 | 36.6 min | 1.0× |
> | 2 | 20.9 min | 1.8× |
> | 4 | 16.6 min | 2.2× |
>
> $G=2$ achieves nearly identical accuracy to $G=1$ (0.1774 vs 0.1764, +0.6%) while reducing memory by 17% and time by 1.7×. $G=4$ offers 48% memory reduction and 2.2× faster time-to-accuracy with 4.9% accuracy trade-off. Beyond $G=8$, speed saturates (~3×) while accuracy degrades sharply.
>
> ---
>
> ### Response to Q5 (35pu):
> The code URL in our paper is a clickable hyperlink embedded in the PDF—clicking it navigates to our anonymous repository. The repository provides preprocessed data, execution instructions to reproduce all main results, and required package versions. If the hyperlink does not work, access directly at: https://anonymous.4open.science/r/EqGINO/

---

> > ### Author Rebuttal · Reviewer_35pu · 2026-04-02
> >
> > Thank you for your reply, which helped me resolve the concern.

---

> > > ### Author Response · Authors · 2026-04-03
> > >
> > > Dear Reviewer 35pu,
> > >
> > > Thank you for confirming that your concerns have been fully resolved. We truly appreciate your time and constructive feedback throughout the discussion period.
> > >
> > > Best regards,
> > > The Authors

---

### Official Review · Reviewer_w9ed · 2026-03-08

**Soundness:** 3
**Presentation:** 4
**Significance:** 3
**Originality:** 3
**Overall Recommendation:** 5
**Confidence:** 5

**Summary:**

The paper introduces EqGINO -- an equivariant version of GINO. The latter is a neural operator architecture which uses Encoder-Processor-Decoder structure. Encoder interpolates from irregular data grid to a regular latent grid where FNO is applied to learn hidden features and then Decoder maps those back to the original irregular grid. The authors use equivariant convolution (Equivariant Graph Neural Operator) to do the interpolation to regular grid and introduce a constraint on weight matrices used in the latent FNO to achieve equivariance to discrete symmetries of the system (Orbit-based Weight Sharing).

**Compliance With Llm Reviewing Policy:**

Affirmed.

**Final Justification:**

I think this is a strong paper and the conference would benefit from having it presented. The authors addressed my concerns regarding the comparison to SOTA approaches and shown that their model performs better in OOD scenarios. I maintain my score (Accept) but raise my confidence (4 -> 5).

**Key Questions For Authors:**

1. The framework is designed for featureless point clouds, but in CFD there could be scalar features (pressure) and vector features (wind velocity). Those would need special treatment, and the method does not straighforwadly allows it.

__Q__: How straightforward is it to use EqGINO for dynamical tasks? What about vector features? Do you have any plans on validating your method on such problems?

2. I have a concern about the local expressivity of GINO and, subsequently, the proposed method. Spectral decomposition requires regular grids, thus, data has to be interpolated to them from native grids. If a geometry is non-uniform, or points have varying density, the interpolation is essentially spatial downsampling which might lose high-frequency details and also introduce aliasing. The latter would be specifically problematic for dynamical tasks.

__Q__: Do you think spatial downsampling might be an issue? If the application is reasonably high-scale, resolving high-frequency features would require finer latent grids, which intuitively would hinder scalability.

3. Another concern is again from GINO itself: it is quite slow to build the graph at the pre-processing step. The computational overhead is quite substantial which is evident from table 3 where both GINO and EqGINO are order of magnitude slower than baselines in both training and inference.

__Q__: Did you profile the method, where are the bottlenecks currently? If it is in preprocessing, is there a way to accelerate it?

4. My only concern regarding the experiments is the choice of baselines that is somewhat limited. Transolver is claimed to be state-of-the-art, while there are methods that surpass it. Specifically for CFD tasks, Erwin [1] and AB-UPT [2] are state-of-the-art approaches. Since the authors themselves stress the importance of global receptive field -- exactly the motivation behind both of those models -- I would expect them to be in the baseline set.

__Q__: How does the model compare against state-of-the-art neural PDE models such as Erwin and AB-UPT?

[1] Erwin: A Tree-based Hierarchical Transformer for Large-scale Physical Systems, Maksim Zhdanov, Max Welling, and Jan-Willem van de Meent, ICML 2025
[2] AB-UPT: Scaling Neural CFD Surrogates for High- Fidelity Automotive Aerodynamics Simulations via Anchored- Branched Universal Physics Transformers, Bleeker, M.J., Dorfer, M., Kronlachner, T., Sonnleitner, R., Alkin, B., & Brandstetter, J., Physics Transformers. Trans. Mach. Learn. Res., 2025.

**Limitations:**

yes

**Strengths And Weaknesses:**

## Presentation
- The presentation is of very high quality.
- The paper is well written and is easy to follow.
- Visualizations are crisp, intuitive, and helpful.

## Soundness
- The theoretical analysis is sound.
- The experimental setup is fair and rigorous.

## Significance
- The problem of consistency of Neural Operators is important, especially for applications where no canonical orientation is provided.
- Equivariance is often a critical property in physical simulations and authors adapt a well-tested architecture to satisfy it.
- The method demonstrates state-of-the-art performance on CFD tasks.

## Originality
- The method is novel and original.
- Authors analyzed where GINO violates equivariance and developed an efficient solution that proved to work in experiments.

---

> ### Author Rebuttal · Authors · 2026-03-31
>
> We greatly appreciate the reviewer's detailed and valuable comments.
>
> ### Response to Q1 (w9ed)
> We appreciate the question, but we note that EqGINO already handles both scalar and vector input features in our experiments. As detailed in Apx Table 2, scalar features (e.g., force magnitude) are directly incorporated as node attributes, while vector features (e.g., inlet velocity, external force) are converted into rotation-invariant quantities such as the dot product with the surface normal (e.g., $\mathbf{u}_{in} \cdot \mathbf{n}$ for AhmedBody/ShapeNetCar, $\mathbf{F} \cdot \mathbf{n}$ for DeepJEB). This preserves equivariance while retaining physical information from vector inputs.
> Regarding dynamical tasks, extending EqGINO to time-dependent problems is a promising future direction. Spatial equivariance would carry over to each time step; the main challenge is coupling temporal evolution with spectral processing.
>
> ---
>
> ### Response to Q2 (w9ed):
>
> This is a valid observation. However, the EqGNO encoder avoids naive spatial downsampling by computing a learnable kernel integration. The MLP-parameterized kernel implicitly encodes sub-grid high-frequency details into high-dimensional latent features rather than smoothing them out. Quadrature weights $\mu_j$ account for varying point densities, providing robustness against aliasing.
>
> Regarding the resolution trade-off, finer grids capture sharper spatial variations, and enriched latent vectors encode geometric patterns with higher fidelity. As Figure 7c confirms, increasing $K$ yields higher prediction accuracy. EqGINO's $O(K)$ parameter complexity requires substantially less memory than standard spectral layers, allowing reasonably dense grids within practical limits.
>
> ---
>
> ### Response to Q3 (w9ed):
>
> Regarding the computational overhead, profiling confirms the primary bottleneck lies in the GNO graph construction. Although we optimize the native $O(N^2)$ complexity to $O(N d r^3)$ with voxelization, hash-based search, and Nyström approximation, radius-based neighbor searches on irregular point clouds fundamentally demand higher overhead than standard grid operations.
>
> We plan to explore an adaptive strategy integrating hierarchical spatial partitioning of Ball Trees [1] with error-driven refinement principles of AMR [2]. Erwin [1] structures point cloud data for fast regional retrieval, whereas AMR [2] dictates where to concentrate computational power based on physical saliency. This dynamically adjusts the search depth according to local geometric complexity, aggregating coarse-grained features in far-field regions and descending to leaf nodes only where physically necessary. We expect this to reduce total FLOPs significantly while preserving the discretization-convergent nature of the operator.
>
> [1] Zhdanov et al., "Erwin: A tree-based hierarchical transformer for large-scale physical systems", ICML 2025.
> [2] Berger & Oliger, "Adaptive mesh refinement for hyperbolic partial differential equations", J. Comput. Phys. 1984.
>
> ---
>
> ### Response to Q4 (w9ed)
>
> We thank the reviewer for suggesting these baselines. We additionally benchmarked Erwin (ICML 2025) and AB-UPT (TMLR 2025) on two representative tasks under both in-distribution and zero-shot rotation settings.
>
> **ShapeNetCar (Pressure):**
>
> | Model | In-dist | Zero-shot (90°) |
> |:---|:---:|:---:|
> | Erwin | 0.1140 | 1.4400 |
> | AB-UPT | 0.0986 | 1.8413 |
> | Transolver | 0.1194 | 1.6632 |
> | GINO | 0.1610 | 1.4950 |
> | EqGINO (ours) | 0.1773 | 0.1773 |
>
> **AhmedBody (Omega):**
>
> | Model | In-dist | Zero-shot (90°) |
> |:---|:---:|:---:|
> | Erwin | 0.0417 | 0.0427 |
> | AB-UPT | 0.0226 | 0.0497 |
> | Transolver | 0.0159 | 0.0469 |
> | GINO | 0.0146 | 0.0385 |
> | EqGINO (ours) | 0.0137 | 0.0137 |
>
> On ShapeNetCar, EqGINO trails in canonical performance. To guarantee equivariance, EqGINO cannot use absolute coordinate features, and the magnitude of this trade-off depends on how strongly coordinates correlate with the target physical quantity. ShapeNetCar simulates all samples with a fixed inlet velocity of 20 m/s in the same direction—the only variation is geometry shape—so the x-coordinate directly encodes the front-to-back pressure gradient, making coordinates a powerful shortcut. On AhmedBody (Omega), which varies inlet velocity across samples (~20–55 m/s) and involves more complex physics, coordinate shortcuts are ineffective, and EqGINO achieves the best in-distribution performance among all models.
>
> Under rotation, all non-equivariant baselines degrade substantially. Notably, AB-UPT exhibits the strongest coordinate dependency (0.0986 → 1.8413 on ShapeNetCar), consistent with its heavy reliance on absolute positional features. EqGINO is the only model that maintains identical performance across orientations by construction.
>
> Due to time constraints, we present two representative tasks here. If accepted, we will include the full comparison across all datasets and tasks with accompanying analysis in the revised manuscript.

---

> > ### Author Rebuttal · Reviewer_w9ed · 2026-04-02
> >
> > I am grateful to the authors for their thoughtful response, and I am happy to see honest engagement with other models. I hope the additional baselines will make it into the final version of the paper as they showcase that right inductive biases allow outperforming strong computer vision inspired models when tested in out-of-distribution scenarios.
> >
> > I maintain my score as Accept but raise my confidence.

---

> > > ### Author Response · Authors · 2026-04-02
> > >
> > > Dear Reviewer w9ed,
> > >
> > > We sincerely thank the reviewer for the encouraging feedback and for raising the confidence score. Your constructive suggestions have greatly helped strengthen our work.
> > >
> > > Best regards,
> > > The Authors

---

### Official Review · Reviewer_XePo · 2026-03-08

**Soundness:** 3
**Presentation:** 3
**Significance:** 3
**Originality:** 3
**Overall Recommendation:** 4
**Confidence:** 3

**Summary:**

This paper introduces a novel neural network framework, named EqGINO, designed to address the trade-off between SE(3)-equivariance and computational efficency in 3D PDE simulations. EqGINO integrates the structural advantages of GINO with the global spectral modeling capability of FNO. By innovatively introducing an orbit-based weight-sharing mechanism, it reduces the parameter complexity of spectral convolution from $O(K^3)$ to $O(K)$ while strictly preserving equivariance to rotations and translations. The model comprises two main modules: EqGNO, which handles irregular geometric inputs, and EqFNO, which performs equivariant spectral learning on regular grids. Experimental validation on multple 3D physical simulation datasets (e.g., AhmedBody, ShapeNetCar, and DeepJEB) demonstrates that EqGINO achieves significant advantages in equivariant generalization, zero-shot rotation robustness, and computational efficiency.

**Compliance With Llm Reviewing Policy:**

Affirmed.

**Final Justification:**

Thank the authors for their response, which addressed my concerns. I maintain my positive recommendation.

**Key Questions For Authors:**

1. In the absence of prior knowledge such as external forces or inlet velocity, can EqGINO achieve equivariance, or are there alternative solutions?

2. When the model encounters a continuous rotation that is not within the discrete octahedral group, what changes occur in its internal representation, and how does this affect the results?

**Limitations:**

I did not find a separate section discussing limitations. The authors are suggested to discuss the content mentioned in Weaknesses and Questions in the revised version.

**Strengths And Weaknesses:**

**Strengths**

1. The design of Orbit-based Weight Sharing and Full-FFT is novel, providing clear insights.

2. The experimental design is comprehensive, covering multiple physical domains such as fluid and solid mechanics, and includes comparisons with various baselines like GNN, PointNet, and Transformer. Additionally, extensive visualization results offer intuitive explanations.

3. While achieving SOTA performance, it does not introduce significant additional computational overhead, as demonstrated in Appendix D.

**Weaknesses**

1. The method is limited to discrete equivariance, and it remains unclear whether it has the potential to be generalized to continuous groups such as SE(3).

2. The construction of the local basis relies on prior knowledge such as external forces or inlet velocity, which may make it difficult to apply in scenarios where such information is insufficient.

---

> ### Author Rebuttal · Authors · 2026-03-31
>
> We thank the reviewer for the thoughtful and constructive feedback.
>
> ---
>
> ### Response to W1 (XePo):
>
> While our *orbit-based weight sharing* targets continuous SO(3) equivariance theoretically, the uniform grid discretization required for the FFT limits exact mappings in practice.
>
> Continuous rotations move coordinates off the discrete grid, requiring spatial interpolation that introduces artifacts and spectral distortion. In contrast, rotations that map grid points exactly back to the lattice avoid this step, preserving exact equivariance.
>
> Despite this limit, our experiments demonstrate that the isotropic weight prior provides a strong structural bias, making EqGINO highly robust to continuous geometric variations. We will discuss this discretization limit in the future work section of the revised manuscript.
>
> ---
>
> ### Response to W2 and Q1 (XePo):
>
> For scalar predictions (e.g., pressure, von Mises stress), our model guarantees equivariance without a local basis. For vector predictions like deflection, the absence of a known canonical pose makes weight conjugation infeasible (Appendix G, *Remark A.3.*). Thus, the local basis is the most appropriate alternative when an external physical vector exists. Where such vectors are absent, we can construct an intrinsic local frame depending solely on intrinsic properties such as normal vectors and principal curvatures (Cohen et al., 2019; Boscaini et al., 2016).
>
> [1] Cohen, Taco, et al. "Gauge equivariant convolutional networks and the icosahedral CNN." *International conference on Machine learning.* PMLR, 2019.
>
> [2] Boscaini, Davide, et al. "Learning shape correspondence with anisotropic convolutional neural networks." *Advances in neural information processing systems* 29 (2016).
>
> ---
>
> ### Response to Q2 (XePo):
> Each orbit $\mathcal{O}_r$ is assigned an independent learnable weight $w_r$, with no coupling between orbits—even those with close radii $r$. Therefore, increasing $K$ adds more independently parameterized orbits, improving canonical expressivity, but does not help continuous-angle interpolation, since orbits with close $\|\mathbf{k}\|_2$ values can still learn arbitrarily different weights. What matters for continuous generalization is that orbits with close radii learn *similar* weight values. To induce this, we apply rotation augmentation during training: at each iteration, the input geometry and its associated physical quantities are randomly rotated to an arbitrary orientation, while the total number of training samples remains unchanged. This encourages orbits with close radii to converge to similar weights in a data-driven manner.
>
> We illustrate this in Figure R1: https://anonymous.4open.science/r/EqGINO/figures/weights_visualization.jpg
>
> *Figure R1: Each square = a frequency mode's learnable weight (color = weight value). (a) Vanilla FNO: no structure. (b) EqGINO w/o aug: same orbit shares weights, but neighbors differ. (c) EqGINO w/ aug: neighboring orbits learn similar weights, enabling interpolation.*
>
> When the input is rotated, the Fourier coefficients rotate correspondingly in the spectral domain (Lemma 4.1). A 90° rotation maps a coefficient exactly onto another grid point (blue circle)—this works for both (b) and (c). A 22.5° rotation maps it *between* grid points (red circle), requiring interpolation from neighbors. In (b), neighboring orbits have unrelated weights, leading to unintended interpolation. In (c), augmentation has induced smoothness, enabling accurate interpolation.
>
> This predicts: (1) $K$↑ without augmentation (Figure R1b) improves canonical but not continuous, and (2) $K$↑ with augmentation (Figure R1c) improves both. We verify on ShapeNetCar:
>
> **Without augmentation** (Figure R1b: close radii $\not\Rightarrow$ close weights):
>
> | $K$ | 0° (Canonical) | 22.5° (Continuous) |
> |:---:|:---:|:---:|
> | 24 | 0.1800 | 0.7982 |
> | 32 | 0.1878 | 0.7801 |
> | 40 | 0.1773 | 0.8018 |
> | 48 | 0.1782 | 0.7971 |
> | 56 | 0.1690 | 0.7661 |
> | *K:24→56* | ↓6.1% | ↓4.0% |
>
> **With augmentation** (Figure R1c: close radii $\Rightarrow$ close weights):
>
> | $K$ | 0° (Canonical) | 22.5° (Continuous) |
> |:---:|:---:|:---:|
> | 24 | 0.1777 | 0.1832 |
> | 32 | 0.1594 | 0.1666 |
> | 40 | 0.1513 | 0.1577 |
> | 48 | 0.1435 | 0.1472 |
> | 56 | 0.1397 | 0.1455 |
> | *K:24→56* | ↓21.4% | ↓20.6% |
>
> *All values are relative $L_2$ error.*
>
> Both predictions are confirmed. **Grid refinement alone is insufficient; the key is that orbits with close radii learn close weights.** As future work, enforcing this smoothness architecturally (e.g., parameterizing orbit weights as a smooth function of $\|\mathbf{k}\|_2$) could provide continuous equivariance guarantees without augmentation.
>
> ---
>
> ### Response to Limitations:
> If accepted, we will add an explicit Limitations section consolidating points such as the discrete-to-continuous gap in a structured manner.

---

> > ### Author Rebuttal · Reviewer_XePo · 2026-04-01
> >
> > Thank the authors for their response, which addressed my concerns. I maintain my positive recommendation.

---

> > > ### Author Response · Authors · 2026-04-01
> > >
> > > Dear Reviewer XePo,
> > >
> > > Thank you for confirming that your concerns have been fully resolved. We truly appreciate your time and constructive feedback throughout the discussion period.
> > >
> > > Best regards,
> > > The Authors

---

### Official Review · Reviewer_iL68 · 2026-03-09

**Soundness:** 3
**Presentation:** 4
**Significance:** 3
**Originality:** 3
**Overall Recommendation:** 5
**Confidence:** 4

**Summary:**

The paper introduces EqGINO, an equivariant extension of GINO [1] for 3D PDEs. EqGINO consists of EqGNO, an equivariant instantiation of GNO to process unstructured point clouds, and EqFNO, an extension of FNO for SE(3) equivariance that processes inputs on structured grids. Compared to prior approaches, EqFNO leverages a novel orbit-based weight sharing mechanism in combination with a local basis to achieve equivariance. Like GINO, EqGINO uses EqGNO to map the unstructured point cloud to a structured latent grid, which is processed by EqFNO. The model is evaluated on 3 benchmark problems under different settings (e.g., in-distribution and zero-shot).

[1] Li, Z. et al. (2023) “Geometry-informed neural operator for large-scale 3D PDEs”

**Compliance With Llm Reviewing Policy:**

Affirmed.

**Final Justification:**

EqGINO provides a valuable equivariant extension of GINO, and the proposed approach has the potential to impact the SciML community. The paper is supported by comprehensive experiments and theoretical analyses, which strengthen its contributions. The rebuttal further clarified the remaining points and satisfactorily resolved my concerns. In sum, I view the paper as a solid contribution and recommend acceptance.

**Key Questions For Authors:**

- What is the GPU memory of EqGINO compared to recent models like Transolver?
- Which errors does Transolver achieve if scaled to 4.5M parameters?
- How does the neighborhood radius of EqGNO influence the model's performance?

**Limitations:**

The paper adequately addresses the limitations of EqGINO. For instance, EqGINO has only exact equivariance for the octahedral rotation group and the computational costs are higher compared to baselines due to the graph construction of EqGNO.

**Strengths And Weaknesses:**

**Strengths**
- The paper is well written, and the derivation of the architecture is clear to understand and contains all the required details.
- The theoretical analyses provide valuable insights and support the design choices made for the proposed model.
- EqGINO is well motivated and addresses the gap of including equivariance into neural PDE solvers.
- A novel orbit-based weight sharing mechanism implements equivariance and, additionally, reduces the parameter count.
- The framework is modular, allowing EqGNO and EqFNO to be used independently for different tasks.
- EqGNO achieves significantly lower errors in the equivariant setting compared to (non-)equivariant baselines.

**Weaknesses**
- While EqGINO performs well in the equivariant settings (zero-shot and continuous), it seems that the equivariant extensions sometimes hurt the performance for in-distribution settings (e.g., 0.1610 for GINO compared to 0.1878 for EqGINO on ShapeNet).
- The method relies on intrinsic geometric features, which may limit expressivity compared to coordinate-based models in some settings.
- The computational costs (time and FLOPS) are significantly higher compared to most of the baselines.
- Transolver has only 1.5M parameters compared to 4.1M for EqGINO accordingly to Table 3 in Appendix D. More parameters for Transolver could potentially lead to even lower errors in the in-distribution setting.
- Highlighting the last row in Table 3 in Appendix D could be misunderstood as highlighting the best values.

---

> ### Author Rebuttal · Authors · 2026-03-31
>
> We sincerely thank the reviewer for the thorough and constructive feedback. We address each point below.
>
> ### Response to W1 (iL68):
> To guarantee equivariance, EqGINO excludes absolute coordinates, which limits expressivity. The magnitude of this trade-off depends on how strongly coordinates correlate with the target quantity—and ShapeNetCar is the dataset where this correlation is strongest.
> ShapeNetCar uses a fixed inlet velocity (20 m/s) across all samples, so the x-coordinate directly encodes the pressure gradient, making it a powerful shortcut. AhmedBody varies inlet velocity (~20–55 m/s) and DeepJEB has four load directions, so coordinate shortcuts are ineffective—and EqGINO matches or outperforms GINO on these datasets (Table 1a).
> Figure 6 confirms this: when rotated 180°, GINO predicts high pressure on the rear bumper simply because it occupies the former front bumper position—memorizing coordinates rather than physics.
>
> ---
>
> ### Response to W2 (iL68):
> We agree this restricts the learnable function space. However, this restriction enables EqGINO to learn the underlying physical law rather than coordinate-dependent shortcuts. As shown in W1 and Figure 6, coordinate memorization contradicts the coordinate-invariance of PDEs and fails under rotation, despite producing low in-distribution error.
>
> ---
>
> ### Response to W3 (iL68):
> Please refer to **[Response to Q3 (w9ed)](https://openreview.net/forum?id=43QdwsbcNM&noteId=V44olmzool)** : "Computational Overhead"
>
> ---
>
> ### Response to W4 and Q2 (iL68):
>
> We scaled Transolver along four axes (width, depth, slice count, and balanced) to match EqGINO's parameter count (~4.1M). Our baseline Transolver configuration (1.5M) was already optimized by referencing the original Transolver paper's recommended hyperparameters. We observed that larger configurations converge more slowly, so we additionally trained all variants for 200 epochs (2× the original schedule).
>
> | Config | Hidden | Layers | Slices | Params | Epochs | Rel $L_2$ |
> |:---|:---:|:---:|:---:|:---:|:---:|:---:|
> | Transolver (baseline) | 128 | 8 | 64 | 1.5M | 100 | 0.01585 |
> | A: Width | 216 | 8 | 64 | 4.2M | 100 | 0.01837 |
> | B: Depth | 128 | 22 | 64 | 4.1M | 100 | 0.03904 |
> | C: Slice+Depth | 128 | 21 | 128 | 4.1M | 100 | 0.03953 |
> | D: Balanced | 192 | 10 | 64 | 4.2M | 100 | 0.03010 |
> | A: Width | 216 | 8 | 64 | 4.2M | **200** | 0.01259 |
> | B: Depth | 128 | 22 | 64 | 4.1M | **200** | 0.01932 |
> | C: Slice+Depth | 128 | 21 | 128 | 4.1M | **200** | 0.01878 |
> | D: Balanced | 192 | 10 | 64 | 4.2M | **200** | 0.01582 |
> | EqGINO (ours) | - | - | - | 4.1M | 100 | 0.0137 |
>
> At 100 epochs, all scaled Transolver variants perform *worse* than the baseline 1.5M configuration, confirming that simply increasing parameters does not directly translate to better performance—larger models require longer training to converge. At 200 epochs, the best variant (Config A, width scaling) reaches 0.01259, which is comparable to EqGINO's 0.0137 achieved in only 100 epochs.
>
> ---
>
> ### Response to W5 (iL68):
> We will revise Table 3 to use bold only for best values, removing the row highlighting.
>
> ---
>
>
> ### Response to Q1 (iL68):
> | Model | Params | Inf. (ms) | Train (ms) | GPU (MB) | GFLOPs |
> |:---|:---:|:---:|:---:|:---:|:---:|
> | Transolver (ICML'24) | 1.5M | 9.1 | 14.6 | 436 | 11.1 |
> | AB-UPT (TMLR'25) | 1.7M | 7.9 | 8.3 | 884 | 12.0 |
> | Erwin (ICML'25) | 5.0M | 20.6 | 24.1 | 2,318 | 45.4 |
> | GINO (NeurIPS'23) | 285M | 63.7 | 64.6 | 11,707 | 153.1 |
> | EqGINO (ours) | 4.1M | 52.6 | 57.7 | 6,600 | 85.7 |
>
> Compared to GINO, EqGINO reduces memory by **44%**, parameters by **98.6%**, and GFLOPs by **44%**. The remaining gap vs. lightweight models stems from 3D Fourier layer training ($K^3$ grid across $L$ layers). We note that Transolver, AB-UPT, and Erwin all degrade under rotation; EqGINO is the only model combining global receptive field with guaranteed equivariance. For accuracy comparisons, see **[Response to Q4 (w9ed)](https://openreview.net/forum?id=43QdwsbcNM&noteId=V44olmzool)**.
>
>
> ---
>
> ### Response to Q3 (iL68):
> We ablated GNO radius $r \in$ { 0.05, 0.075, 0.1, 0.125, 0.15, 0.175, 0.2 } on ShapeNetCar:
>
> | $r$ | 0.05 | 0.075 | 0.1 | 0.125 | 0.15 | 0.175 | 0.2 |
> |:---|:---:|:---:|:---:|:---:|:---:|:---:|:---:|
> | Rel $L_2$ | .1887 | .1812 | .1793 | .1756 | .1774 | .1779 | .1744 |
>
> Too-small radii ($r < 0.1$) aggregate too few mesh points per grid node, yielding sparse features. For $r \in [0.1, 0.2]$, error varies within only ~2.8%, demonstrating that performance is robust to the choice of radius. While $r=0.2$ achieves the lowest error, larger radii increase neighbor count and GNO cost ($O(Ndr^3)$). We selected $r=0.15$ as it achieves performance within 1.7% of the best while keeping computational overhead manageable.

---

> > ### Author Rebuttal · Reviewer_iL68 · 2026-04-01
> >
> > Thank you for your detailed response and the additional results, which have addressed most of my concerns. I am raising my score accordingly.

---

> > > ### Author Response · Authors · 2026-04-01
> > >
> > > Dear Reviewer iL68,
> > >
> > > We sincerely appreciate your engagement. Your feedback helped us meaningfully improve the manuscript, and we are grateful that the revisions addressed your concerns. Thank you for your time and thoughtful evaluation!
> > >
> > > Best regards,
> > > The Authors

---

### Decision · Program_Chairs · 2026-04-30

**Decision:**

Accept (regular)

**Comment:**

This paper proposes EqGINO, an equivariant extension of GINO for 3D PDE surrogate modeling, combining an equivariant geometric encoder with an equivariant Fourier module based on orbit-wise spectral weight sharing. Reviewers agreed that the paper addresses an important problem, is clearly presented, and is supported by substantial experiments.

The main strengths are  technical clarity, careful design, and a solid evaluation that supports claims. The zero-shot rotation experiments and qualitative evidence are convincing, and the rebuttal helped clarify tradeoffs, baselines, and intended scope.

Remaining concerns are more about scope: whether equivariance holds only for the discrete symmetries, while continuous are only empirical. There is also an expressivity tradeoff: the equivariant bias can hurt in-distribution performance when absolute coordinates play a role.

I think that this is a strong and useful paper. The idea is interesting, execution convincing, and the experiments strong enough for acceptance. I do think the paper could be more explicit about the distinction between exact discrete equivariance and empirical continuous-rotation robustness and the various tradeoffs.

I recommend acceptance.

For the camera-ready, the authors should clarify the scope of the guarantees, discuss the expressivity tradeoff more plainly, and tighten the discussion of limitations and computational overhead.